# SecCodePRM: A Process Reward Model for Code Security

**Weichen Yu** [1]  **Ravi Mangal** [2]  **Yinyi Luo** [1]  **Kai Hu** [1]  **Jingxuan He** [3]  **Corina S. Păsăreanu** [1]  **Matt Fredrikson** [1]

## Abstract

Large Language Models are rapidly becoming core components of modern software development workflows, yet ensuring code security remains challenging. Existing vulnerability detection pipelines either rely on static analyzers or use LLM/GNN-based detectors trained with coarse program-level supervision. Both families often require complete context, provide sparse end-of-completion feedback, and can degrade as code length grows, making them ill-suited for real-time, prefix-level assessment during interactive coding and streaming generation. We propose **SecCodePRM**, a security-oriented process reward model that assigns a **context-aware**, **step-level** security score along a code trajectory. To train the model, we derive step-level supervision labels from static analyzers and expert annotations, allowing the model to attend more precisely to fine-grained regions associated with inter-procedural vulnerabilities. SecCodePRM has three applications: full-code vulnerability detection (VD), partial-code VD, and secure code generation (CG). For VD, SecCodePRM uses risk-sensitive aggregation that emphasizes high-risk steps; for CG, SecCodePRM supports inference-time scaling by ranking candidate continuations and favoring higher cumulative reward. This design yields dense, real-time feedback that scales to long-horizon generation. Empirically, SecCodePRM outperforms prior approaches in all three settings, while preserving code functional correctness, suggesting improved security without a safety–utility tradeoff. Code is available at SecCodePRM

[1]Carnegie Mellon University [2]Colorado State University [3]University of California, Berkeley. Correspondence to: Weichen Yu <wyu3@andrew.cmu.edu>.

*Proceedings of the 43ʳᵈ International Conference on Machine Learning*, Seoul, South Korea. PMLR 306, 2026. Copyright 2026 by the author(s).

## 1. Introduction

Large Language Models (LLMs) have demonstrated remarkable capabilities in code generation, significantly enhancing developer productivity and helping in real-time software development and code design (Chen, 2021; Copet et al., 2025; Jiang et al., 2024; Zheng et al., 2023). As these models become increasingly integrated into development workflows, ensuring the security of LLM-generated code, including vulnerable code detection and secure code generation, has emerged as a critical research priority (Huynh & Lin, 2025; Dai et al., 2025).

Prior work on vulnerable code detection largely falls into three categories: (i) purely static-analysis-based approaches (Lipp et al., 2022; Avgustinov et al., 2016), (ii) hybrids that couple LLM reasoning with static analysis tools (Sandoval et al., 2023; Pearce et al., 2025), and (iii) learned models trained on labeled full programs, including LLM-based and GNN-based methods (Wang et al., 2023b; Nguyen et al., 2022; Qiu et al., 2024) as well as hybrid architectures (Wang et al., 2020; Yang et al., 2024; Lekssays et al., 2025). Despite steady progress, these lines of work exhibit important limitations. Static analysis tools struggle in settings where only partial code is available, often require full program context to achieve reasonable accuracy, and can be time costly at scale (Mohsin et al., 2024). Meanwhile, many LLM/GNN-based detectors are trained with coarse, program-level binary supervision over full code, which provides limited pressure to attend to fine-grained vulnerable cross-functional/cross-file regions; moreover, their effectiveness can degrade as code length and context grow.

To address these limitations, we introduce **SecCodePRM**, a process reward modeling approach that bridges the gap between how human experts spot vulnerabilities early, as the code is being written, and how automated tools typically require complete programs. Rather than waiting for a full code block to run static analyses or LLM/GNN-based detectors, SecCodePRM evaluates partial code trajectories step by step, assigning a context-aware safety score as each step of the code is written or generated. **During training**, we construct step-level safety labels from static analyzers (e.g., CodeQL) or human annotations, and the model learns to detect the moment a vulnerability is introduced by conditioning each step's prediction on the preceding prefix.

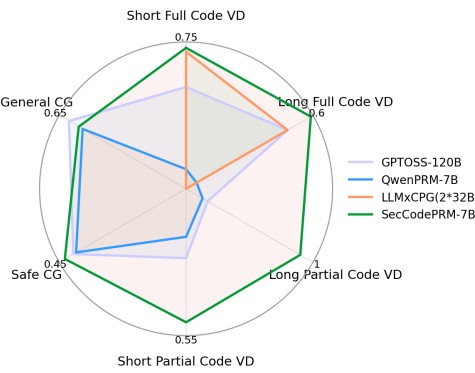

*Figure 1.* Performance comparison on vulnerability detection (VD) and code generation (CG). SecCodePRM consistently outperforms prior approaches across three core settings: VD on complete programs, VD on partial code prefixes, and safety-guided CG, compared to off-the-shelf LLMs, PRMs, and SOTA VD methods, and without the expense of general CG tradeoff.

**SecCodePRM's Diverse Applications.** SecCodePRM can tackle both complete and partial code vulnerability detection (VD) and safe code generation (CG). **During inference**, for VD, step rewards are aggregated with a risk-sensitive weighting that emphasizes low-safety steps; for safe CG, the same reward signal enables inference-time scaling, ranking multiple candidate continuations and allocating compute toward trajectories with higher cumulative safety. This yields earlier, denser feedback than full-completion methods, reducing training overhead while remaining aligned with expert security triage.

We emphasize the importance of task *partial-code VD*, which is a timely need for **real-time evaluation during interactive coding or streaming generation**. SecCodePRM is well-suited for this task by assigning a context-aware safety score at each generation step. As code generation scales from function-level to repo-level, SecCodePRM **provides dense, immediate reward signals throughout the trajectory**—supporting both safer inference-time search (by ranking candidate continuations) and more efficient optimization in settings such as reinforcement learning, where full-rollout reward computation becomes increasingly expensive.

As summarized in fig. 1, SecCodePRM consistently outperforms prior approaches across three core settings: VD on complete programs, VD on partial code prefixes, and safety-guided CG. While off-the-shelf large language models, (e.g., GPT-oss-120B) perform strongly on general-purpose CG, they often lack reliable capability for secure code detection and secure code generation. Existing state-of-the-art VD approaches such as LLMxCPG (Lekssays et al., 2025) typically rely on full context and may fail to identify vulnerabilities from partial code snippets. General off-the-shelf process reward models (e.g., QwenPRM) are not trained with security-oriented objectives and thus provide limited guidance for producing or validating secure

code. The proposed SecCodePRM consistently improves performance in both VD and security-aware CG. Importantly, these gains do not come at the expense of general code generation quality—functionality-oriented generation remains unchanged and is occasionally improved—suggesting that SecCodePRM strengthens security without introducing the typical safety–utility tradeoff.

## 2. Related Work

**LLMs in Safe Code Generation.** While LLMs and autonomous LLM agents have demonstrated remarkable proficiency in code generation tasks (Jimenez et al., 2024; Dong et al., 2025; Zhuo et al., 2025), the security posture of their outputs remains a critical concern. Recent empirical evidence suggests that the safety of LLM-generated code is often suboptimal (Wang et al., 2023a; Mohsin et al., 2024; Chong et al., 2024; Ramírez et al., 2024), frequently introducing exploitable vulnerabilities. This has catalyzed a growing body of research focused on the rigorous evaluation of code safety (Liu et al., 2024; Wang et al., 2024a; Dai et al., 2025; Mou et al., 2025), with particular emphasis on high-stakes domains such as web applications (Dora et al., 2025) and autonomous driving systems (Nouri et al., 2025). To bridge this safety gap, several methodologies have emerged for secure code generation. These include the integration of LLMs with traditional static analysis tools to verify outputs (Dolcetti et al., 2024; Nunez et al., 2024), specialized prompting strategies (Tony et al., 2024), and the use of constrained decoding to enforce security properties during the inference process (Fu et al., 2024; Zhang et al., 2024). Furthermore, optimization-based approaches, such as soft prompt tuning, have been proposed to steer models toward generating more secure implementations (He & Vechev, 2023; Nazzal et al., 2024).

**Code Vulnerability Detection.** Recent literature has extensively explored the utility of LLMs for automated vulnerability detection (Ding et al., 2024). A prevalent approach involves supervised fine-tuning, where a binary classification head is integrated atop an LLM (Fu et al., 2022; Kim et al., 2022; Sun et al., 2023; Zhou et al., 2024; Du et al., 2024a). To capture structural dependencies, several studies have hybridized LLMs with Graph Neural Networks (GNNs) to improve predictive accuracy (Lu et al., 2024; Yang et al., 2024; Liu et al., 2025). Beyond fine-tuning, prompting-based strategies have gained traction (Fu et al., 2023; Khare et al., 2023), leveraging techniques such as Chain-of-Thought (CoT) reasoning (Ullah et al., 2024), Retrieval-Augmented Generation (RAG) (Du et al., 2024b), and specialized frameworks targeting specific flaw classes like Use-Before-Initialization (Li et al., 2024) or smart contract vulnerabilities (Sun et al., 2024). Despite these methodological advances, empirical evidence suggests that both fine-tuning and prompting-based paradigms continue to struggle with

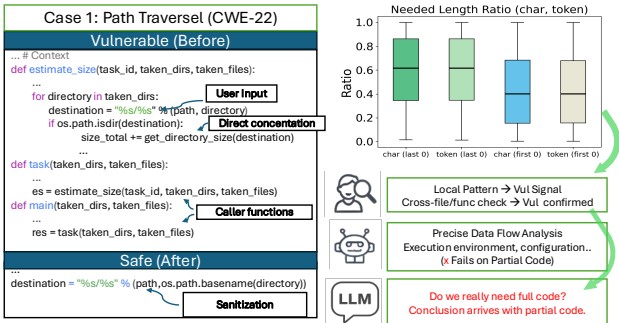

*Figure 2.* Human vs. Automated Vulnerability Detection. Left: A CWE example, we use labels on the vulnerability code, and their transitive closure of caller functions. Right: Box plot showing expert humans identify flaws using only $\sim 60\%$ of code tokens from the beginning. Flow analysis usually fails on partial code and requires extra information.

the inherent complexities of vulnerability detection (Ding et al., 2024; Hannan et al., 2025), leaving a significant performance gap compared to expert human analysis.

To our knowledge, existing vulnerability detection methodologies are trained and validated exclusively on full code implementations. This training paradigm fundamentally limits their applicability and accuracy when applied to partial code vulnerability detection scenarios, creating a critical gap between current capabilities and the requirements of modern LLM-assisted development environments.

# 3. SecCodePRM

## 3.1. The Semantic Gap Between Human Expertise and Automated Vulnerability Detection

A fundamental asymmetry exists between expert human reasoning and automated security evaluation. We measure the token required from the beginning of the code to localize all vulnerable code, including the transitive closure of caller functions that invoke the vulnerable function. In the box plot, as shown in fig. 2, our analysis of vulnerability detection patterns reveals that human experts can identify security flaws, such as path traversal (CWE-22), by observing only 40% to 60% of the code tokens. This suggests that seasoned practitioners could recognize the "telltale signature" of a vulnerability at around 40% tokens, such as the dangerous co-occurrence of unsanitized user input and then confirm that the vulnerability exists at around 60% tokens, e.g. a sensitive API sink, before the full block is completed.

In contrast, current detection methods typically require a complete program to perform static analysis or LLM-based pattern matching. For example, requires execution environment, compile-time macros, configuration flags etc to reach a final decision. This "full-completion" requirement imposes **two significant limitations**: first, it **ignores the strong**

**predictive signals** available in partial code prefixes; second, it creates **a computational bottleneck** during training (e.g., Reinforcement Learning), where generating a full rollout for a single reward signal is increasingly cost-prohibitive as code generation lengths grow from function-level to repo-level.

To address these inefficiencies, we propose a partial-code vulnerability detection approach, which can also be used for safe code generation with inference time scaling. By shifting from global program verification to partial detection, our method allows for an earlier "reflexive judgment" during the generation process. This design choice is expected to provide a dense, immediate reward signal that significantly reduces training overhead while maintaining high alignment with expert security triage.

## 3.2. Problem Setup

We address two coupled tasks: (i) **Vulnerability Detection**, where a model must classify whether a (potentially partial) code trajectory $\tau$ contains security flaws, and (ii) **Safe Code Generation**, where we leverage the detector to guide a generator via *inference-time scaling*. A trajectory $\tau = (s_1, ..., s_T)$ consists of $T$ reasoning or code steps. Our objective is to learn a Process Reward Model (PRM), $r_\theta$, that assigns a safety score to each $s_t$. We assume access to a dataset $\mathcal{D} = \left( (\tau^{(i)}, y_{1:T^{(i)}}^{(i)}) \right)_{i=1}^N$ where $y_t$ are step-level safety annotations derived from static analysis tools (e.g., CodeQL) or human experts. Unlike prior work requiring full project compilation, our model must remain robust to **partial code contexts** common in real-time generation.

## 3.3. Secure Code Process Reward Modeling

**Code Steps.** The SecCodePRM operates at the granularity of discrete code steps, rather than complete code sequences. This allows the model to pinpoint the exact moment a security vulnerability is introduced in the reasoning or generation process. For inference, per step is separated by a separator (e.g., \n\n). Each time this separator is generated, the PRM evaluates the current partial trajectory to produce a process reward. For training, the entire code is first separated into steps using the same separator followed by a further data cleaning step that processes the code's Abstract Syntax Tree (AST) and either merges or separates some of the steps to align better with logical boundaries.

**Context-Aware Code Safety.** Unlike traditional classifiers that either classify code snippets in isolation or classify the entire code as a whole, our PRM is trained to evaluate each step $s_t$ within the context of the preceding trajectory $\tau_{<t} = (s_1, ..., s_{t-1})$. During training, we provide the model with the complete code trajectory. We inject a training signal at every step separator token. For a trajectory with $T$ steps, the model's last layer is a classification head, and

produces a sequence of secure probabilities $\mathbf{h}_1, ..., \mathbf{h}_T$, $h \in R^2$ corresponding to each separator.

**Step-level Margin Scoring.** For each step $s_t$, we compute a reward $r_t$ based on the logit margin between safe (+) and vulnerable (-) classes:

$$r_t = \text{softmax}(\mathbf{h}_t)^{(+)} - \text{softmax}(\mathbf{h}_t)^{(-)}. \qquad (1)$$

This margin $r_t \in [-1,1]$ acts as a localized "safety signal." A sharp drop in $r_t$ between step $t-1$ and $t$ typically indicates the introduction of a security flaw.

**Reweighted Aggregation for Detection.** To detect vulnerabilities in partial or full code, we aggregate step rewards using a temperature-scaled weighting that emphasizes "risk-heavy" steps:

$$R = \sum_{i=1}^{T} w_i \cdot r_i, \quad w_i = \frac{\exp(-r_i/\tau)}{\sum_{j=1}^{T}\exp(-r_j/\tau)} \qquad (2)$$

where lower rewards (higher risk) receive exponentially higher weights.

**Inference-Time Scaling for Generation.** To ensure safe code generation, we use the PRM to scale inference compute. Given a generator $G$, we produce $K$ candidate steps at each time $t$. The PRM $r_\theta$ ranks these candidates using **Advantage-based Selection.** We define the advantage of a step as $A_t = r_t - \mathbb{E}[r_t]$, where $\mathbb{E}[r_t]$ is estimated by an average of the whole batch. During search, we prioritize trajectories that maximize the cumulative reweighted reward $R$. This allows the model to abandon trajectories that lead to unsafe states, effectively shifting compute from low-safety to high-safety paths.

**Training Objective and Optimization.** The PRM is trained using a standard cross-entropy objective, with a weighting factor for balancing the minority "vulnerable" class to prevent the PRM from being overly optimistic.:

$$\mathcal{L}(\theta) = -\sum_{t=1}^{T}\log w_y P_\theta(y_t \mid s_t, \tau_{<t}), \qquad (3)$$

where $w_y$ is based on its ground truth label. $y_t$ is the ground-truth safety label for step $t$ conditioned on all prior context. This ensures the reward $r_t$ reflects not just the local syntax of $s_t$, but its semantic safety given the existing variable definitions and state.

**Comparison with Other Pipelines.** The SecCodePRM architecture offers several key advantages over traditional LLM + Graph (Yang et al., 2024; Du et al., 2024a) or LLM + ToolUse (Lekssays et al., 2025) approaches shown in the fig. 3. First, while other methods typically require full code blocks for analysis, SecCodePRM is flexible enough

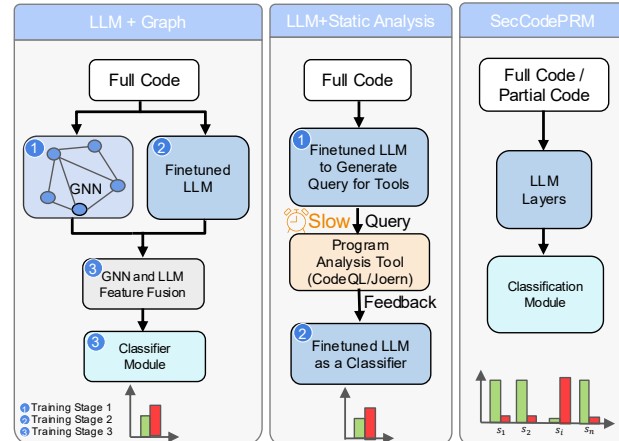

*Figure 3.* Pipeline Comparison.

to process both partial and completed code trajectories. Second, unlike the "Slow" multi-step process involving external program analysis tools (like CodeQL or Joern), SecCodePRM operates as an end-to-end system, making it significantly faster during inference and highly suitable for providing real-time feedback as an RL training reward. Third, SecCodePRM benefits from training simplicity; whereas the graph-based approach requires three distinct training stages to fuse GNN and LLM features, SecCodePRM is streamlined into a one-stage training process.

## 4. Dataset Construction and Refinement

To train and evaluate the PRM for vulnerability detection, we develop a systematic data construction pipeline that transforms raw function-level / repo-level code into granular, step-aligned supervision signals. The pipeline in fig. 4 focuses on paired data: contrasting vulnerable code with its corresponding security patches to identify specific vulnerable spans.

**Granular Step Segmentation.** Unlike traditional full-code-level classification, the PRM requires a step-wise decomposition of the input to provide dense supervision. We define a "step" as a logical block of code delimited by double newlines (\n\n). Formally, for a given function $F$, we decompose it into a sequence of steps $S = \{s_1, s_2, ..., s_n\}$. This decomposition allows the model to assign rewards to specific segments of the code, facilitating more interpretable and localized vulnerability localization.

**Compact and Structural Merging via Heuristic Patterns.** Raw code segmentation often yields artifacts that contains noise or unnecessary contents. We define a filtering function $\phi(s)$ that removes any step $s$ if it matches a set of "no-op" patterns, including empty strings, whitespace-only lines, or comment-only blocks (e.g., /**/, //).

To maintain the integrity of logical blocks (e.g., ensuring a `return` statement is not detached from the preceding logic), we implement a merging strategy based on suffix and prefix

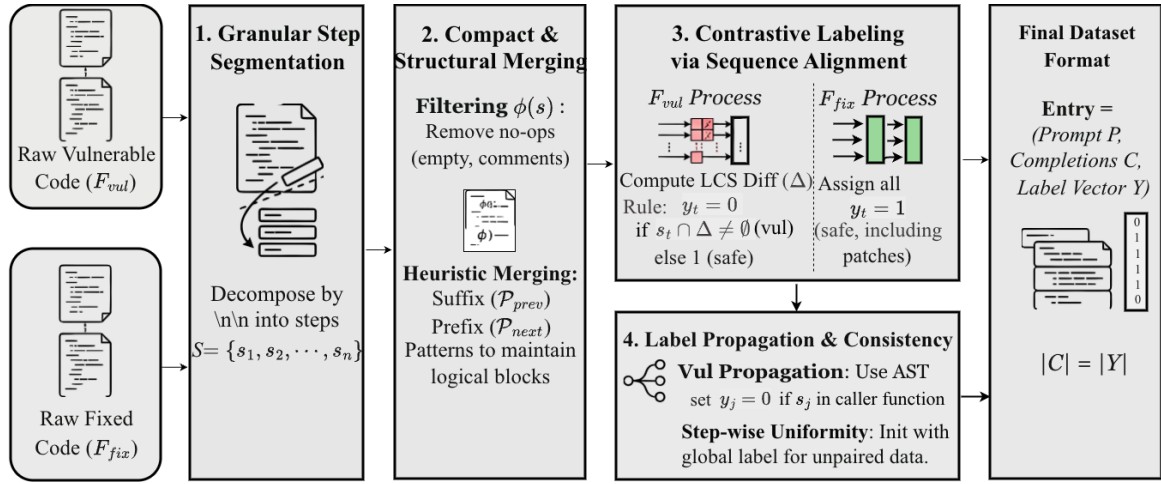

*Figure 4.* Dataset construction, cleaning, labeling pipeline.

patterns. We define two sets of boundary heuristics:

- **Suffix Patterns** ($\mathcal{P}_{prev}$)**:** Statements that logically terminate a block, such as `return 0;`, `break;`, or `#endif`. If $s_i \in \mathcal{P}_{prev}$ and $Label(s_i) = Label(s_{i-1})$, we merge the steps: $s_{i-1} \leftarrow s_{i-1} \oplus s_i$, where $\oplus$ refers to string concatenation.

- **Prefix Patterns** ($\mathcal{P}_{next}$)**:** Variable declarations or header definitions (e.g., `register ssize_t i;`) that provide context for the following line. If $s_i \in \mathcal{P}_{next}$ and $Label(s_i) = Label(s_{i+1})$, we merge: $s_{i+1} \leftarrow s_i \oplus s_{i+1}$.

This process ensures that the PRM evaluates more complete semantic blocks rather than fragmented lines.

**Contrastive Labeling via Sequence Alignment.** The core of supervised signal derives from the structural differences between vulnerable steps ($s_{vul}$) and their fixed counterparts ($s_{fix}$). We employ a sequence alignment algorithm based on the Gestalt Pattern Matching (Giuliano et al., 1961) to compute the longest common subsequences (LCS).

Let $\Delta$ be the set of identified differences (diffs) between $F_{vul}$ and $F_{fix}$. For the vulnerable function $F_{vul}$, we assign a step-level label $y_t$ to each step $s_t$ according to the following rule:

$$y_t = \begin{cases} 0 \ (\text{vulnerable}), & \text{if } s_t \cap \Delta \neq \emptyset \\ 1 \ (\text{secure}), & \text{otherwise} \end{cases} \quad (4)$$

For the fixed function $F_{fix}$, all segments—including the newly introduced patches—are assigned a positive label ($y_t = 1$), as they represent the resolved and secure state of the code. This contrastive approach ensures that the model learns to penalize specific lines responsible for a vulnerability rather than the invariant boilerplate code.

**Label Propagation and Consistency.** To handle datasets where only sequence-level (i.e., full=code-level) labels are available, or alignment is ambiguous, we implement two heuristic refinement strategies: 1) **Vulnerability Propagation:** For known vulnerable samples, we identify the first point of failure $k$. Following the assumption that a single vulnerability corrupts the transitive closure of caller functions, if a step $s_k$ is identified as the root cause, the caller function steps $s_j > s_i$ are also assigned a negative label ($y_j = 0$). We use Abstract Syntax Tree (AST) for labeling the caller functions which are after the callee function. 2) **Step-wise Uniformity:** In cases where fine-grained diffs are unavailable (unpaired data), we initialize the step-level labels with the global label of the function. This provides a baseline supervision signal that the PRM can iteratively refine during training.

**Dataset Format.** Each entry in the resulting dataset consists of a formatted prompt $P$ in natural language, a sequence of completions $C = \{c_1, c_2, ..., c_n\}$ which could be natural language and code, and a corresponding label vector $Y \in \{0,1\}^n$, where $|C| = |Y|$. This structure enables the model to learn the transition from secure code to vulnerable code at any point within the function.

### 4.1. Training Datasets Analysis

We analyze the following commonly used public vulnerability detection datasets: **BigVul** (Fan et al., 2020), a commonly used but noisy C and C++ dataset; **SVEN** (He & Vechev, 2023), a small high-quality pairwise C and Python code dataset with human-annotated labels; **PrimeVul** (Ding et al., 2024), a large-scale, automatically collected C and C++ dataset with chronological data splitting strategies to mitigate data leakage issues; **ReposVul** (Wang et al., 2024b), a repo-level automatically collected C and C++ dataset with multi-granularity information; and **PreciseBugs** (He et al., 2023), an automated precise repo-level bug collection from open-source projects, using a bug tracker and a bug injector. PreciseBugs contains six programming languages.

We first analyze their attributes, as in table 13. **Example V:S** means the ratio of the number of vulnerable examples over the number of secure examples. **Token V:S** means the ratio of the average token number of each example, **Char V:S** means the ratio of the average character count of each example, **Step V:S** means the ratio of the average step count of each example, separated by the predefined separator. **Step$_v$ Ratio** means the average vulnerable step percentage in vulnerable examples, where **Step$_a$ Ratio** means the average vulnerable step percentage in all examples. CWE types are in the appendix.

Here are the analysis of the datasets, as in table 13: 1) **Imbalanced Ratio:** In the example level, *BigVul*, *PrimeVul (Unpaired)*, and *ReposVul* are heavily skewed toward secure samples, with vulnerability ratios below 10%. Even at the vulnerable examples, the actually vulnerable steps only take a very small proportion, ranging from 7% to 27% at the step level. And the average vulnerable step in all steps go down to less than 1%. 2) **Length Difference**. From function level to repo-level, the average number of tokens increases from hundreds to 20K, which will increase difficulties for detection.

**Token Length Hacking.** We notice that the BigVul and the PrimeVul Unpaired datasets have very different number of tokens for vulnerable examples and secure examples: on average, the vulnerable examples can be 3 times longer than the secure examples. This attribute could be hacked by LLMs and thus the model may depend only on length instead of on real security features. Thus, we exclude these data from training.

## 5. Experiments

**Implementation and Training Details.** Detection Threshold is 0.5. Scaling Strategy: $N = 10$ for SVEN and N=20 for CWEval following their settings. All experiments were conducted on a single compute node equipped with NVIDIA 4*80GB GPUs (e.g., A100 or H100). To manage memory efficiency and accelerate training, we integrate DeepSpeed ZeRO-2. We use the **Qwen2.5-Coder-7B-Instruct** as our base transformer backbone, and added a classification head. We first finetune only the classification head and then the full parameter. **Hyperparameters and Optimization.** We train the model for 3 epochs using a constant learning rate of $1 \times 10^{-4}$ with the Adam optimizer ($\beta_2 = 0.95$). No warmup ratio was applied. Training was performed in Bfloat16 precision to maintain numerical stability while reducing memory overhead. Detailed hyperparameters are summarized in table 10.

**Tools on Different Datasets.** Following previous works, different benchmark uses different tools/evaluation methods, we clarify as follows:

- **Full-code vulnerability detection:** no external tools are used; evaluation is based on ground-truth labels.

- **Partial-code vulnerability detection:** no external tools

*Table 1.* Vulnerability Detection on full code samples from SVEN. For a fair comparison, we follow previous work (Lekssays et al., 2025), and the reported numbers are on CWE-125, CWE-190, CWE-416 and CWE-476 on SVEN.

| | VulSim | VulBERTA CNN | VulBERTA MLP | ReGVD | LLMx CPG | LLM Prompting | SecCodePRM |
|---|---|---|---|---|---|---|---|
| Acc | 33 | 50 | 50 | 51 | 60 | 50.9–54.5 | **72** |
| F1 | 31 | 44 | 43 | 55 | 70 | – | **72** |

are used; evaluation is based on ground-truth labels.

- **Secure code generation:** on **SVEN**, security is evaluated with **CodeQL**; on **CWEval**, security is evaluated with **unit tests** following the benchmark setup.

- **General code generation:** on **LiveCodeBench**, evaluation is based on **unit-test passing**.

### 5.1. Vulnerability Detection on Full Code Samples

**Evaluation Metrics.** We evaluate the performance of *SecCodePRM* using standard classification metrics: accuracy, precision, recall, and the F1 score. For the PrimeVul dataset, we adopt the more rigorous pair-wise evaluation framework proposed by (Ding et al., 2024), including Pair-wise Correct Prediction (P-C), Pair-wise Vulnerable (P-V), Pair-wise Benign (P-B), and Pair-wise Reversed (P-R). Notably, the P-C metric serves as a strict upper bound on model reliability, as it requires the model to correctly classify both the vulnerable and the corresponding patched version of a code snippet.

**SVEN.** As in table 1, baseline methods are VulSim (Shimmi et al., 2024), which uses similarity of embeddings, VulBERTA (Hanif & Maffeis, 2022) uses pretraining, ReGVD (Nguyen et al., 2022) uses GNNs, and LLMxCPG (Lekssays et al., 2025) finetunes two 32B Qwen model. SecCodePRM outperforms other methods, which shows that process-level supervision is more effective at capturing function-level semantics than simply increasing model scale or using graph-based structural priors on function-level human-annotated benchmarks. Especially, SecCodePRM surpasses LLMxCPG, which uses a static analysis tool (Joern) and larger models in the pipeline, by a margin of 12%, showing that the static analysis tools have limitation on accuracy.

**PrimeVul.** On PrimeVul in table 2 and table 3, the results demonstrate the most substantial gap between our method and existing baselines. Baseline methods include the code language models (CodeT5 (Wang et al., 2021), CodeBERT (Feng et al., 2020), UnixCoder (Guo et al., 2022), CodeGen2.5 (Nijkamp et al., 2023), StarCoder2 (Lozhkov et al., 2024)), and the security specific methods (LLMxCPG), The improvement on PrimeVul is significant, with a more than 30% gap in accuracy compared to SOTA methods. Critically, our model achieves a P-V and P-R rate of 0%, indicating that it almost never incorrectly classifies both samples in a pair as vulnerable, nor does it inversely predict labels. This level of

*Table 2.* VD on full code samples from PrimeVul Paired (Pairwise Metrics following (Ding et al., 2024)), described in section B.5.

| Model / Method | PC↑ | PV↓ | PB↓ | PR↓ |
|---|---|---|---|---|
| CodeT5 | 1.06 | 12.94 | 84.75 | 1.24 |
| CodeBERT | 0.35 | 1.95 | 86.17 | 0.71 |
| UniXcoder | 1.60 | 12.06 | 85.11 | 1.24 |
| StarCoder2 | 2.30 | 8.16 | 88.30 | 1.24 |
| CodeGen2.5 | 3.01 | 10.82 | 84.22 | 1.95 |
| Qwen2.5-PRM-7B | 3.41 | 82.07 | 11.34 | 3.17 |
| GPT-3.5 (Two-shot) | 5.67 | 13.83 | 77.84 | 2.66 |
| GPT-3.5 (CoT) | 6.21 | 4.79 | 83.51 | 5.50 |
| GPT-3.5 (Fine-tune) | 1.24 | 5.32 | 90.96 | 2.48 |
| GPT-4 (Two-shot) | 5.14 | 71.63 | 21.45 | 1.77 |
| GPT-4 (CoT) | 12.94 | 54.26 | 24.47 | 8.33 |
| RANDOM GUESS | 22.70 | 26.24 | 26.42 | 24.65 |
| *SecCodePRM* | **93.66** | **0** | **6.34** | **0** |

*Table 3.* Vulnerability Detection on full code samples from Prime-Vul Paired (Classification Metrics following (Lekssays et al., 2025)).

| Model / Method | Acc↑ | F1↑ |
|---|---|---|
| Qwen2.5-PRM-7B | 50.12 | 63.15 |
| LLMxCPG | 72.50 | 62.06 |
| *SecCodePRM* | **96.83** | **96.73** |

consistency suggests that *SecCodePRM* has learned a robust representation of the underlying vulnerability rather than relying on superficial syntax shortcuts.

*Table 4.* VD on full code completions from PreciseBugs.

| | F1 | Precision | Recall |
|---|---|---|---|
| Random | 0.29 | 0.20 | 0.50 |
| CodeLlama | 0.22 | 0.16 | 0.35 |
| LineVul | 0.31 | 0.43 | 0.25 |
| MSIVD | 0.48 | 0.40 | 0.57 |
| SecCodePRM | **0.59** | **0.53** | **0.67** |

**PreciseBugs.** This dataset has extremely long contexts, containing complex code examples exceeding 20,000 tokens. As shown in table 4, baseline methods includes code models CodeLlama (Roziere et al., 2023), LineVul (Fu & Tantithamthavorn, 2022)—a transformer based line-level vulnerability detection, and MSIVD (Yang et al., 2024) which combines embeddings from an LLM and a GNN to make final decision. Our proposed method outperforms others by a large gap, showing significant resilience to sequence length.

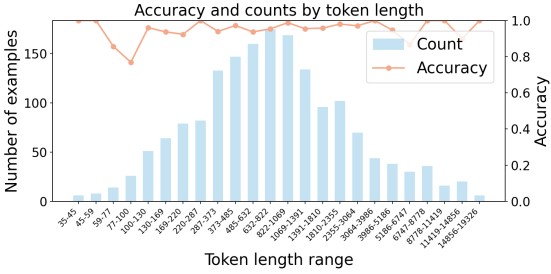

*Figure 5.* Token length and accuracy distribution on PrimeVul.

**Accuracy and Token Length.** To further investigate this resilience, we analyze the relationship between code length

and model performance on PrimeVul. As illustrated in fig. 5 and fig. 8, the accuracy of *SecCodePRM* remains stable even as token counts increase significantly. This confirms that the process-based reward modeling effectively mitigates the "lost in the middle" or signal-decay issues common in traditional sequence classification models, enabling reliable vulnerability detection in large-scale codebases.

**PRM vs. FRM Ablation.** To isolate the contribution of process-level supervision is essential to substantiate the claim that the observed improvements stem from the intrinsic advantage of step-level reward modeling rather than from additional exposure to vulnerability-labeled data, we conduct a controlled ablation in which a Final Reward Model (FRM), trained exclusively with full-trajectory supervision, is compared against SecCodePRM under otherwise identical training conditions. The results are reported in table 5. As shown in table 5, SecCodePRM substantially outperforms the FRM baseline on both metrics, improving accuracy and F1. This large margin indicates that the gains cannot be attributed to task-specific exposure alone, and instead provide direct empirical evidence for the unique contribution of process-level reward modeling. Moreover, we note that SecCodePRM additionally supports partial VD, a capability that the FRM formulation is fundamentally unable to provide, further underscoring the methodological advantage of step-level supervision.

*Table 5.* Ablation comparing the Final Reward Model (FRM) and SecCodePRM under otherwise identical training conditions.

| Model / Method | Acc | F1↑ |
|---|---|---|
| FRM | 72.28 | 61.65 |
| *SecCodePRM* | **96.83** | **96.73** |

### 5.2. Vulnerability Detection on Partial Code Samples

**Evaluated Metrics.** We evaluate performance using *step-wise* accuracy and F1-score across four diverse benchmarks: SVEN, PrimeVul, ReposVul, and PreciseBugs, where the ground truth label are collected using previous data pipeline.

**Comparative Baselines.** We benchmark our method against prompting a comprehensive suite of state-of-the-art LLMs, ranging from specialized coding models to massive reasoning architectures. The prompt used are in section B.6. These include the Qwen-Coder (QC) series (7B to 32B), Llama-4-Scout (17B), Codestral (22B), Gemma-3 (12B), and the GPT-oss-120B series (spanning low, medium, and high reasoning configurations).

**Results.** From the experimental results summarized in fig. 6 and table 11 we can draw the following conclusions. **1)** We observe that most general-purpose LLMs maintain high accuracy but exhibit poor F1-scores. For instance, while massive models like GPT-oss-120B demonstrate high

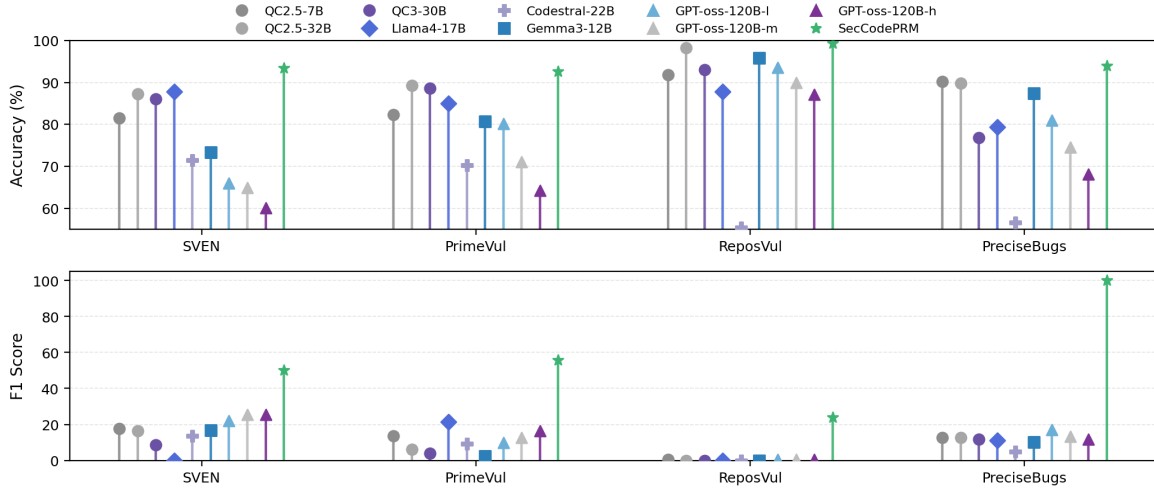

*Figure 6.* Partial vulnerability detection results across 4 benchmarks.

classification accuracy, they consistently fail to surpass the 30% F1 threshold on most benchmarks. This suggests a tendency toward "conservative" predictions, where models **frequently misclassify vulnerable code as benign**, thereby inflating accuracy at the expense of security reliability. **2)** Contrary to the standard scaling laws observed in general benchmarks, increased parameter counts do not linearly correlate with better vulnerability identification. This indicates that the nuances of partial code vulnerability are not inherently captured by massive-scale general pre-training. **3)** SecCodePRM (7B) **achieves state-of-the-art results, outperforming models nearly 17 times its size**. Our method achieves a near-perfect F1-score of 100.00 on PreciseBugs and maintains robust F1 performance across all other benchmarks. By excelling in both precision and recall, SecCodePRM demonstrates that process-level reward modeling provides a significantly more effective inductive bias for security tasks than raw parameter scaling or general instruction tuning. A case study is in section B.9.

**Train with Propagation, Evaluate without Propagation.** Since label propagation is employed during dataset preprocessing, a natural question is whether the reported gains are an artifact of this heuristic rather than a genuine improvement in model capability. To disentangle these effects, we train on the propagated dataset but evaluate exclusively against ground-truth labels, as reported in table 6. The results demonstrate that SecCodePRM's performance is largely invariant to the choice of evaluation protocol: ground-truth evaluation yields nearly identical scores on **SVEN** and **PrimeVul**, and even slightly outperforms the propagation-based protocol on **ReposVul** and **PreciseBugs**. These findings indicate that the improvements attributable to SecCodePRM are not contingent on the propagation heuristic and remain robust under the stricter ground-truth evaluation regime.

*Table 6.* Comparison of evaluation with propagation vs. evaluation on ground-truth labels.

| | **SVEN** | | **PrimeVul** | |
| --- | --- | --- | --- | --- |
| | F1 | Acc | F1 | Acc |
| Eval on Propagation | 50.17 | 93.45 | 55.71 | 92.57 |
| Eval on Ground Truth | 49.52 | 92.85 | 55.33 | 92.52 |
| | **ReposVul** | | **PreciseBugs** | |
| | F1 | Acc | F1 | Acc |
| Eval on Propagation | 23.79 | 99.33 | 90.00 | 93.94 |
| Eval on Ground Truth | 24.36 | 99.35 | 92.86 | 94.41 |

### 5.3. Secure Code Generation with Inference Time Scaling

We evaluate the efficacy of PRM in inference-time scaling for secure programming, using a base generator (QC2.5-7B/QC2.5-32B) and rank them using baseline PRM (Qwen/Qwen2.5-Math-PRM-7B) and SecCodePRM (SCPRM) to select the top-$k$ candidates. This approach allows us to decouple the generation capability from the verification capability, testing whether our proposed method can effectively steer the model toward secure yet functional code.

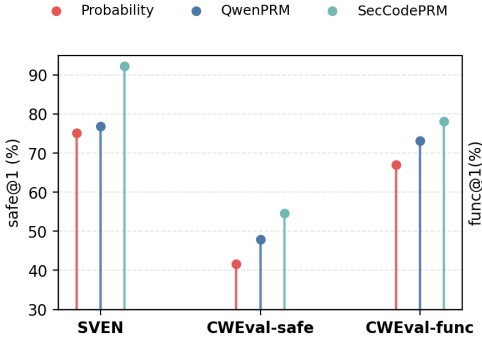

*Figure 7.* Performance on safe ratio (SR@1, safe@1) and function correctness on SVEN and CWEval safe code generation benchmark.

**Security-Centric Evaluation on SVEN.** We first evaluate on the SVEN benchmark (He & Vechev, 2023), which utilizes hand-selected contexts and CodeQL-based automated verification. We report the **Security Rate (SR@$k$)**, defined as the percentage of secure programs among valid generations. As illustrated in fig. 7, at SR@1, SecCodePRM yields a significant increase over the random selection baseline (approx. 75%) and outperforms the Qwen/Qwen2.5-Math-PRM-7B. While performance converges at SR@3, and QwenPRM (Qwen/Qwen2.5-Math-PRM-7B) shows slight advantages at SR@5, the primary strength of SecCodePRM lies in its **high-precision ranking for top-1 selection**, which is critical for real-world deployments where only a single suggestion is presented to the developer. Ablation on the reward design is in section B.2.

**Joint Functional and Security Verification on CWEval.** To ensure that security does not come at the expense of utility, we evaluate on CWEval (Peng et al., 2025), which requires code to be both functionally operational (passing unit tests) and vulnerability-free. We report pass@$k$ (functionality) and safe@$k$ (dual requirement of functionality and safety)[1]. Results in fig. 7, table 9 and fig. 10 indicate that both PRM-augmented methods exceed the baseline by a large margin. Notably, SecCodePRM acheives an increasement of 11.09% on func@1 and increase of 12.98% on safe@1. This suggests that SecCodePRM's training allows it to **prioritize the most salient security features without compromising logical correctness**. We attribute this to the data construction, while a safe code data is also the high quality code data.

**Generalizability on LiveCodeBench.** A common failure mode in safety-tuned models is "taxing" general performance. We test this on LiveCodeBench (Jain et al., 2024) to assess general competitive programming proficiency. Metrics are pass@k. As shown in table 7, SCPRM mostly maintains or even improves general functionality. Specifically, on the QC2.5-32B backbone, SCPRM achieves a pass@1 of 55.19%, outperforming both the Probability-based ranking (53.23%) and QwenPRM (Qwen/Qwen2.5-Math-PRM-7B) (53.23%).

Unlike general LLM safety alignment, which often suffers from a "tax" on helpfulness, specialized secure code generation via inference scaling appears to act as a regularizer that enhances general logic. By filtering out buggy or insecure implementations, the PRM effectively selects for higher-quality code overall, suggesting that security and functionality are synergistic dimensions in the coding domain.

---

[1]We adopt the standard pass@k metric, which computes the success rate over k independent samples, rather than best@k. SR@k and safe@k refer to the same metric, though the terminology varies across different benchmarks.

*Table 7.* Results on LiveCodeBench.

| Model | Method | k=1 | k=3 | k=5 |
|---|---|---|---|---|
| QC2.5 -7B | Probability | 40.51 | 45.40 | 47.16 |
| | QwenPRM | **43.44** | **45.99** | 46.97 |
| | SecCodePRM | 41.49 | 45.40 | **47.55** |
| QC2.5 -32B | Probability | 53.23 | 57.93 | **60.27** |
| | QwenPRM | 53.23 | 57.73 | 59.59 |
| | SecCodePRM | **55.19** | **58.12** | 59.69 |

## 6. Conclusion

In this work, we introduced SecCodePRM, a process-based reward modeling approach that bridges the semantic gap between expert human security reasoning and automated detection. By shifting from global program verification to step-wise evaluation, our method successfully identifies "vulnerability signals" in both partial and completed code trajectories. This architecture avoids the multi-stage complexity of traditional graph-based or time consuming tool-assisted pipelines. Furthermore, SecCodePRM enables efficient inference-time scaling for generation, providing dense, real-time reward signals that align closely with the human practitioners and static tools. Our results demonstrate that localized security signals not only improve detection accuracy but also significantly improve secure code generation.

## Acknowledgments

This research is based upon work supported in part by the Office of the Director of National Intelligence (ODNI), Intelligence Advanced Research Projects Activity (IARPA), via 560000C260017. The views and conclusions contained herein are those of the authors and should not be interpreted as necessarily representing the official policies, either expressed or implied, of ODNI, IARPA, or the U.S. Government. The U.S. Government is authorized to reproduce and distribute reprints for governmental purposes notwithstanding any copyright annotation therein.

## Impact Statement

This paper presents work aimed at improving the security of code generated by large language models, a timely concern as LLMs become increasingly integrated into software development workflows. The potential positive societal impact includes reducing exploitable vulnerabilities in production software, thereby enhancing cybersecurity across critical systems in healthcare, finance, and infrastructure. By enabling earlier detection of security flaws during code generation rather than post-hoc analysis, SecCodePRM could help prevent vulnerabilities from reaching deployment.

We acknowledge the dual-use nature of security research: while our method is designed to improve code safety, the

underlying techniques for identifying vulnerable patterns could theoretically be repurposed to craft exploits. However, we believe the defensive benefits substantially outweigh this risk, as the model is specifically trained to flag and avoid vulnerabilities rather than generate them. We do not release trained model weights publicly without appropriate access controls. Additionally, our reliance on existing public vulnerability datasets (with proper chronological splits) mitigates concerns about enabling novel attack vectors. We encourage responsible disclosure practices and recommend that practitioners combine SecCodePRM with established security review processes rather than treating it as a complete replacement for human expertise.

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

## A. Limitations

Our work has limitations that present opportunities for future research. SecCodePRM is trained primarily on C, C++, and Python datasets, and while PreciseBugs includes additional languages (Go, Rust, JVM-based), the generalization to other programming languages and paradigms (e.g., functional languages, domain-specific languages) remains underexplored. Also, our evaluation focuses on function-level and repository-level benchmarks with known CWE categories; the model's effectiveness on zero-day vulnerabilities or novel attack patterns outside the training distribution is uncertain.

## B. Additional Empirical Results

### B.1. More Results on Accuracy vs Token Length

fig. 8 reports the distribution of examples across token-length bins in the PreciseBugs setting together with performance as a function of length. The dataset is dominated by mid-length samples, with the highest counts occurring in the central token ranges and progressively fewer instances at the shortest and longest lengths. Accuracy remains relatively stable across the length spectrum once sufficient data are available, indicating that performance does not substantially degrade with increasing context length in the regime where most samples lie. The pronounced fluctuations in the smallest bins—most notably the sharp dip for very short snippets—are consistent with high variance due to limited support, rather than a systematic length effect. Overall, the result suggests that the method's effectiveness is largely robust to code length, with reliability primarily governed by sample density in each bin.

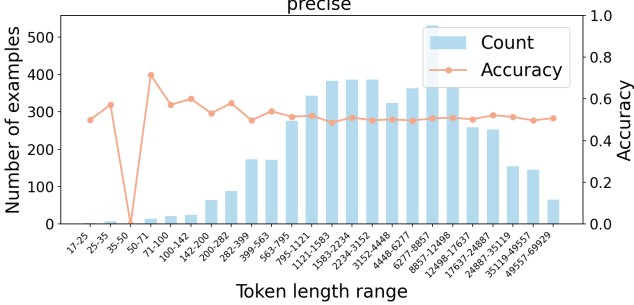

*Figure 8.* Token length and accuracy distribution on PreciseBugs.

### B.2. Ablation on Reward Design

During inference time scaling, we tried different ways for calculating the reward per sample. Here, $r_1$ is defined the advantage of a step as $A_t = r_t - \mathbb{E}[r_t]$, where $\mathbb{E}[r_t]$ is estimated by an average of the whole batch. $r_2$ is defined as the weighted reward from each step, $r_2 = \sum w_t * r_t$, where $w_t = softmax(r_t)$.

*Table 8.* Ablation on per step reward design and aggregation method design. ITS uses Best-of-N, where N=10. UB is the upperbound.

| Method | SR@1 | UB | SR@3 | UB | SR@5 | UB | Avg |
|---|---|---|---|---|---|---|---|
| | 71 | 92 | 69 | 92 | 72 | 88 | 69 |
| softmax+min | 85 | 92 | 74 | 92 | 77 | 92 | 75 |
| last pos+min/binary/ave | 85 | 92 | 79 | 92 | 80 | 92 | 75 |
| softmax+ave | 62 | 92 | 74 | 92 | 72 | 92 | 75 |
| softmax+binary | 85 | 92 | 79 | 92 | 80 | 92 | 75 |

fig. 9 presents an ablation over the reward design on the SVEN safe code generation benchmark, reporting the security ratio SR@(k) for ($k \in 1, 3, 5$). Across all (k), SafeCodePRM yields consistently higher security ratios than the off-the-shelf Qwen-PRM, indicating that security-aligned reward modeling more effectively steers generation toward safe outputs. The sensitivity to (r) differs markedly between the two reward models: SafeCodePRM achieves strong SR@1 at both ($r_1$) and ($r_2$), while QwenPRM shows only modest variation and remains substantially lower at (k=1). As (k) increases, SR@(k) improves overall, reflecting the benefit of sampling multiple candidates; however, SafeCodePRM maintains an advantage under both settings of

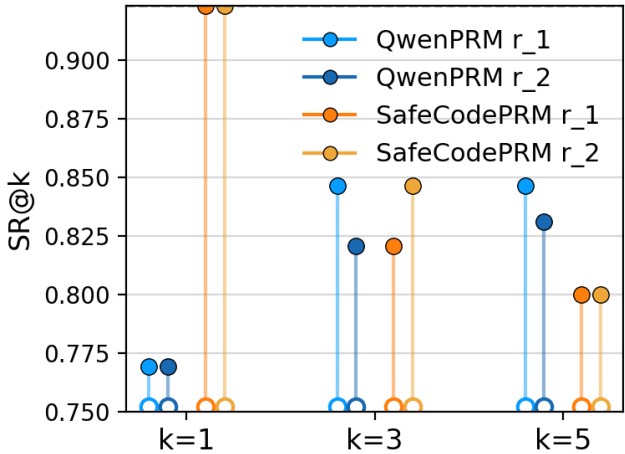

*Figure 9.* Ablation on $r$. Security ratio (SR@k) on SVEN safe code generation benchmark.

(r), suggesting that the gains are not driven by a single hyperparameter choice. Collectively, these results show that the proposed reward model is robust to the choice of (r) and more effective at promoting secure code generation than a general-purpose PRM.

Assume each generated candidate has $T$ steps, and the PRM provides a step score $s_t \in \mathbb{R}$ for step $t$. Let $\sigma(\cdot)$ denote the logistic sigmoid. Let the final aggregated reward be $R$.

**(1) softmax + min.** We compute softmax weights over steps and take a weighted minimum:

$$w_t = \frac{\exp(s_t/\tau)}{\sum_{j=1}^{T}\exp(s_j/\tau)}, \qquad R = \min_{t\in\{1,\ldots,T\}} w_t s_t. \tag{5}$$

**(2) last pos + min / binary / ave.** We define the last-position score as

$$s_{\text{last}} = s_T. \tag{6}$$

The aggregation $R$ is instantiated as follows.

**min:**

$$R = \min_{t\in\{1,\ldots,T\}} s_t. \tag{7}$$

**binary:**

$$R = \mathbb{I}\left[\min_{t\in\{1,\ldots,T\}} s_t \geq 0\right]. \tag{8}$$

**ave:**

$$R = \frac{1}{T}\sum_{t=1}^{T} s_t. \tag{9}$$

(If we explicitly use the last position as the candidate score, we set $R = s_T$.)

**(3) softmax + ave.** Softmax weights followed by a weighted average:

$$w_t = \frac{\exp(s_t/\tau)}{\sum_{j=1}^{T}\exp(s_j/\tau)}, \qquad R = \sum_{t=1}^{T} w_t s_t. \tag{10}$$

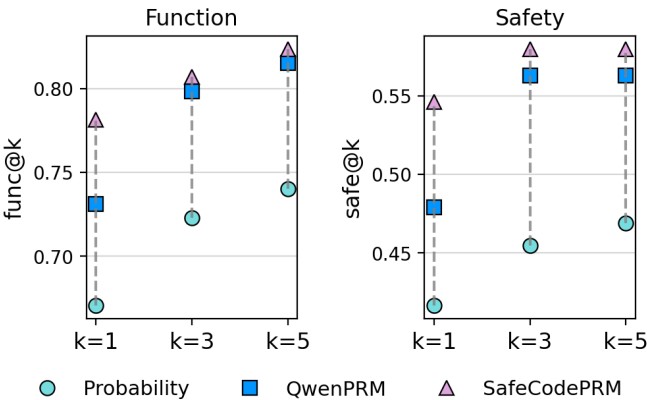

*Figure 10.* Performance on CWEval safe code generation benchmark, func@k and safe@k refers to selecting $k$ candidates from $N=20$ candidates, safe@k requires both functionality correct and safe.

*Table 9.* Results on CWEval func and func-sec metrics.

| Generation Model | Method | func | | | func-sec | | |
|---|---|---|---|---|---|---|---|
| | | @1 | @3 | @5 | @1 | @3 | @5 |
| QC2.5-7B | Probability | 56.81 | 65.83 | 68.77 | 32.65 | 40.21 | 42.97 |
| | QwenPRM | 57.14 | 64.71 | 70.59 | 38.66 | 46.22 | 51.26 |
| | SecCodePRM | 59.66 | 67.22 | 68.91 | 42.86 | 47.06 | 49.58 |
| QC3-30B | Probability | 67.06 | 72.31 | 74.03 | 41.64 | 45.46 | 46.90 |
| | QwenPRM | 73.11 | 79.83 | 81.51 | 47.90 | 56.30 | 56.30 |
| | SecCodePRM | 78.15 | 80.67 | 82.35 | 54.62 | 57.98 | 57.98 |

**(4) softmax + binary.** Softmax weights form an expected score, then we threshold into a binary reward:

$$w_t = \frac{\exp(s_t/\tau)}{\sum_{j=1}^{T}\exp(s_j/\tau)}, \qquad R = \mathbb{I}\left[\sum_{t=1}^{T} w_t s_t \geq 0\right]. \tag{11}$$

table 8 ablates both the per-step reward design and the aggregation strategy used to convert step-level scores into a single candidate-level reward for best-of- N selection (ITS, N=10). Across SR@1/3/5, the baseline exhibits the weakest performance, whereas all alternative designs substantially improve security ratios. Among them, strategies that either (i) apply a softmax weighting over steps and then aggregate (e.g., softmax+min/binary/ave) or (ii) rely on the last-position score combined with simple pooling (last pos+min/binary/ave) achieve the strongest and most consistent SR@k. The reported UB columns indicate that the best achievable security ratio within the candidate pool is high and relatively stable across settings, suggesting that remaining gaps are largely due to the ranking induced by the reward rather than the absence of secure candidates. Overall, the ablation supports that reward shaping at the step level and the choice of aggregation rule materially affect security-aware selection quality.

### B.3. More Results on CWEval

Figure 10 and table 9 compares functionality and safety on the CWEval safe code generation benchmark as a function of the number of selected candidates $k$ (from a pool of $N=20$). The left panel reports func@$k$, measuring the probability that at least one of the top-$k$ candidates is functionally correct, while the right panel reports safe@$k$, which additionally requires the selected candidate to be both correct and secure. Across all $k \in \{1,3,5\}$, both reward-model-based selection strategies outperform probability-based ranking, indicating that explicit reward guidance improves best-of-$N$ selection beyond likelihood alone. In particular, SafeCodePRM consistently achieves the highest func@$k$ and safe@$k$, demonstrating simultaneous gains in correctness and security. As $k$ increases, performance monotonically improves for all methods, reflecting the benefit of evaluating multiple candidates; however, the relative ordering remains stable, suggesting that the improvements stem from better ranking rather than increased sampling alone.

## B.4. Training Parameters

table 10 summarizes the hyperparameter configuration used for PRM training. We adopt a two-stage schedule, training for 3 epochs in stage 1 and 1 additional epoch in stage 2, while decaying the learning rate from $(1\times10^{-4})$ to $(1\times10^{-6})$ to stabilize late-stage optimization. Optimization is performed with Adam $((\beta_1=0.9, \beta_2=0.95))$ and a weight decay of 0.1 to improve generalization. To scale training across hardware, we use a global batch size of $(8\times N_{\text{GPUs}})$ and enable efficient distributed execution with DeepSpeed ZeRO-2. Finally, we set the maximum sequence length to 128,000 tokens to accommodate long-context supervision during PRM learning.

*Table 10.* Hyperparameters for PRM Training

| Hyperparameter | Value |
|---|---|
| Number of Epochs | 3 for stage 1 and 1 for stage 2 |
| Learning Rate | $1\times10^{-4}$ for stage 1 and $1\times10^{-6}$ for stage 2 |
| Optimizer | Adam ($\beta_1=0.9,\beta_2=0.95$) |
| Weight Decay | 0.1 |
| Global Batch Size | $8\times N_{GPUs}$ |
| Max Sequence Length | 128,000 |
| Distributed Strategy | DeepSpeed ZeRO-2 |

## B.5. Evaluation Metric

For PrimeVul, apart from F1 and Acc, the benchmark includes pair-wise metric:

1. Pair-wise Correct Prediction (P-C): The model correctlypredicts the ground-truth labels for both elements of a pair.

2. Pair-wise Vulnerable Prediction (P-V): The model incorrectly predicts both elements of the pair as vulnerable.

3. Pair-wise Benign Prediction (P-B): The model incorrectlypredicts both elements of the pair as benign.

4. Pair-wise Reversed Prediction (P-R): The model incorrectlyand inversely predicts the labels for the pair.

## B.6. Prompts

Prompts for baseline method in partial code vulnerability detection.

> Prompts
>
> "Given the previous code ..., determine whether the current code ... is vulnerable or not. Reason step by step, and answer with Yes or No. Put your answer in the {}"

## B.7. More Results on Partial Vulnerability Detection

table 12 reports results on partial-code vulnerability detection on SVEN across a range of LLMs. Overall, off-the-shelf instruction-tuned models exhibit limited effectiveness in this setting: smaller general-purpose models (e.g., CodeLlama-22B and Gemma3-12B) yield low $F_1$ scores, while QCC variants show particularly poor recall, indicating difficulty in identifying vulnerable snippets without full program context. Larger and more capable models improve precision and accuracy but still struggle to achieve balanced detection; for example, GPT-oss-120B attains higher precision but only moderate recall, resulting in an $F_1$ that remains far from robust. Notably, SafeCodePRM (7B) substantially improves recall and overall detection quality, achieving the best $F_1$ and accuracy among the evaluated systems, which suggests that security-aligned reward modeling provides a stronger signal for vulnerability recognition under partial-context constraints. Finally, the *Invalid* rate varies widely across baselines, highlighting an additional practical limitation of general-purpose prompting for this task and further motivating specialized training and scoring for secure code understanding.

*Table 11.* Comparison of different models on partial code vulnerability detection.

| Model | SVEN | | PrimeVul | | ReposVul | | PreciseBugs | |
|---|---|---|---|---|---|---|---|---|
| | F1 | Acc | F1 | Acc | F1 | Acc | F1 | Acc |
| QC2.5-7B | 17.59 | 81.50 | 13.51 | 82.26 | 0.35 | 91.77 | 12.72 | 90.15 |
| QC2.5-32B | 16.56 | 87.27 | 6.15 | 89.29 | 0.00 | 98.21 | 12.66 | 89.84 |
| QC3-30B | 8.55 | 86.09 | 4.00 | 88.55 | 0.00 | 92.99 | 11.75 | 76.79 |
| Llama4-17B | 0.00 | 87.78 | 21.43 | 85.03 | 0.00 | 87.78 | 10.95 | 79.33 |
| Codestral-22B | 13.57 | 71.45 | 9.30 | 70.31 | 0.00 | 55.34 | 5.00 | 56.65 |
| Gemma3-12B | 16.81 | 73.28 | 2.40 | 80.63 | 0.00 | 95.85 | 10.03 | 87.39 |
| GPT-oss-120B-l | 22.10 | 65.92 | 9.79 | 80.18 | 0.41 | 93.59 | 16.92 | 80.92 |
| GPT-oss-120B-m | 25.33 | 64.93 | 12.53 | 71.04 | 0.39 | 89.90 | 13.35 | 74.56 |
| GPT-oss-120B-h | 25.52 | 60.12 | 16.52 | 64.29 | 0.51 | 87.15 | 11.60 | 68.09 |
| SecCodePRM (7B) | **50.17** | **93.45** | **55.71** | **92.57** | **23.79** | **99.33** | **90.00** | **93.94** |

*Table 12.* Results on partial code vulnerability detection on SVEN.

| Model | Precision | Recall | F1 | Acc | Invalid (%) |
|---|---|---|---|---|---|
| QC2.5-7B | 32.41 | 12.07 | 17.59 | 81.50 | - |
| QC2.5-32B | 22.32 | 13.16 | 16.56 | 87.27 | - |
| QC3-30B | 6.94 | 11.11 | 8.55 | 86.09 | 4.71 |
| Llama4-17B | 0.00 | 0.00 | 0.00 | 87.78 | 82.54 |
| Codestral-22B | 21.43 | 9.93 | 13.57 | 71.45 | 17.10 |
| Gemma3-12B | 26.32 | 12.35 | 16.81 | 73.28 | 8.18 |
| GPT-oss-120B-l | 50.65 | 14.13 | 22.10 | 65.92 | 0.00 |
| GPT-oss-120B-m | 62.34 | 15.89 | 25.33 | 64.93 | 0.00 |
| GPT-oss-120B-h | 71.43 | 15.54 | 25.52 | 60.12 | 0.25 |
| SecCodePRM (7B) | 45.45 | 55.97 | 50.17 | 93.45 | - |

## B.8. Training and Test Dataset Analysis

As in table 13 and table 14, we include the table of training set and test set, including their programming language, CWE type, example numbers, token numbers, char, step and ratios. The datasets SVEN, PreciseBugs, PrimeVul, and ReposVul represent a evolution in the field of automated vulnerability detection and classification. SVEN is a high-quality dataset curated from critical Common Weakness Enumeration (CWE) types, specifically designed to train Large Language Models (LLMs) to distinguish between safe and vulnerable code patterns. PreciseBugs addresses the ambiguity often found in traditional datasets by providing a formal, time-based split of executable bug-fix pairs, ensuring more accurate labeling for research. Moving toward larger scale, PrimeVul offers a comprehensive benchmark with over 220,000 benign and 7,000 vulnerable functions, covering more than 140 CWEs with human-level labeling accuracy. Finally, ReposVul pushes the boundaries further by focusing on repository-level complexities; it encompasses over 6,000 CVE entries across 236 CWE types and four programming languages, specifically targeting "inter-procedural" vulnerabilities that occur across multiple files and functions rather than in isolation. Together, these datasets provide the rigorous, multi-layered data needed to advance the next generation of AI-driven security tools. The PreciseBugs dataset contains the longest samples on a per-instance basis and exhibits the greatest diversity, spanning the widest range of programming languages and the most varied set of CWE categories.

*Table 13.* Train and Test Set Statistics

| Benchmark | Lang | CWE | Split | Example V:S | Token V:S | Char V:S | Step V:S | Step$_v$ | Step$_a$ |
|---|---|---|---|---|---|---|---|---|---|
| BigVul (2020) | C, C++ | [1] | train | $\frac{8700}{150922}$ (5.76%) | $\frac{720}{256}$ | $\frac{2609}{915}$ | $\frac{13}{4}$ | 25.43 | 2.40 |
| | | | test | $\frac{199}{7706}$ (2.52%) | $\frac{590}{244}$ | $\frac{2076}{862}$ | $\frac{10}{4}$ | 26.73 | 1.15 |
| Sven (2023) | C, C++, Python | [2] | train | $\frac{720}{720}$ (50%) | $\frac{828}{843}$ | $\frac{2964}{3022}$ | $\frac{15}{15}$ | 24.20 | 5.99 |
| | | | test | $\frac{83}{83}$ (50%) | $\frac{862}{877}$ | $\frac{3292}{3352}$ | $\frac{14}{14}$ | 21.84 | 5.81 |
| PreciseBugs (2023) | C, C++, Python, Go, JVM, Rust | [3] | train | $\frac{19579}{19579}$ (50%) | $\frac{19383}{19673}$ | $\frac{55219}{55775}$ | $\frac{125}{126}$ | 12.98 | 1.94% |
| | | | test | $\frac{2447}{2448}$ (50%) | $\frac{10491}{10535}$ | $\frac{34312}{34493}$ | $\frac{115}{115}$ | 12.65 | 1.83% |
| PrimeVul (Paired) (2023) | C, C++ | [4] | train | $\frac{7305}{7305}$ (50%) | $\frac{1491}{1528}$ | $\frac{5314}{5454}$ | $\frac{22}{23}$ | 20.17 | 6.90 |
| | | | test | $\frac{854}{854}$ (50%) | $\frac{1492}{1525}$ | $\frac{5296}{5429}$ | $\frac{20}{21}$ | 18.65 | 5.74 |
| PrimeVul (Unpaired) (2023) | C, C++ | [5] | train | $\frac{4862}{170935}$ (2.77%) | $\frac{1508}{342}$ | $\frac{5415}{1179}$ | $\frac{18}{5}$ | – | – |
| | | | test | $\frac{549}{24239}$ (2.21%) | $\frac{1515}{346}$ | $\frac{5360}{1196}$ | $\frac{17}{5}$ | – | – |
| ReposVul (2024) | C, C++ | [6] | train | $\frac{785}{11830}$ (6.63%) | $\frac{12901}{11011}$ | $\frac{43451}{39191}$ | $\frac{185}{154}$ | 7.45 | 0.26 |
| | | | test | $\frac{242}{2914}$ (8.30%) | $\frac{15682}{12816}$ | $\frac{54055}{44784}$ | $\frac{182}{176}$ | 8.97 | 0.29 |

*Table 14.* CWE types in the dataset.

| Dataset | CWE Types |
|---|---|
| SVEN | CWE-022, CWE-078, CWE-079, CWE-089, CWE-125, CWE-190, CWE-416, CWE-476, CWE-787 |
| PreciseBugs | CWE-17, CWE-19, CWE-20, CWE-21, CWE-22, CWE-23, CWE-29, CWE-59, CWE-73, CWE-74, CWE-75, CWE-76, CWE-77, CWE-78, CWE-79, CWE-80, CWE-88, CWE-89, CWE-90, CWE-93, CWE-94, CWE-95, CWE-113, CWE-115, CWE-116, CWE-117, CWE-119, CWE-120, CWE-121, CWE-122, CWE-125, CWE-126, CWE-129, CWE-130, CWE-131, CWE-134, CWE-178, CWE-184, CWE-185, CWE-189, CWE-190, CWE-191, CWE-193, CWE-200, CWE-201, CWE-203, CWE-208, CWE-209, CWE-212, CWE-235, CWE-241, CWE-248, CWE-250, CWE-252, CWE-254, CWE-255, CWE-263, CWE-264, CWE-266, CWE-267, CWE-268, CWE-269, CWE-275, CWE-276, CWE-277, CWE-280, CWE-281, CWE-284, CWE-285, CWE-287, CWE-290, CWE-294, CWE-295, CWE-304, CWE-305, CWE-306, CWE-307, CWE-310, CWE-311, CWE-312, CWE-319, CWE-320, CWE-321, CWE-323, CWE-324, CWE-326, CWE-327, CWE-330, CWE-331, CWE-335, CWE-337, CWE-338, CWE-345, CWE-346, CWE-347, CWE-350, CWE-352, CWE-354, CWE-358, CWE-359, CWE-361, CWE-362, CWE-367, CWE-369, CWE-377, CWE-378, CWE-384, CWE-388, CWE-399, CWE-400, CWE-401, CWE-404, CWE-415, CWE-416, CWE-425, CWE-426, CWE-427, CWE-428, CWE-434, CWE-436, CWE-441, CWE-444, CWE-459, CWE-460, CWE-470, CWE-471, CWE-476, CWE-494, CWE-502, CWE-521, CWE-522, CWE-524, CWE-527, CWE-532, CWE-534, CWE-539, CWE-547, CWE-552, CWE-565, CWE-597, CWE-601, CWE-610, CWE-611, CWE-613, CWE-617, CWE-620, CWE-639, CWE-640, CWE-641, CWE-648, CWE-662, CWE-665, CWE-667, CWE-668, CWE-669, CWE-670, CWE-672, CWE-674, CWE-681, CWE-682, CWE-684, CWE-692, CWE-697, CWE-704, CWE-706, CWE-707, CWE-732, CWE-749, CWE-754, CWE-755, CWE-758, CWE-763, CWE-770, CWE-772, CWE-774, CWE-776, CWE-787, CWE-798, CWE-805, CWE-823, CWE-824, CWE-829, CWE-834, CWE-835, CWE-838, CWE-840, CWE-843, CWE-862, CWE-863, CWE-908, CWE-909, CWE-913, CWE-916, CWE-918, CWE-922, CWE-924, CWE-940, CWE-941, CWE-943, CWE-1004, CWE-1021, CWE-1022, CWE-1077, CWE-1187, CWE-1188, CWE-1220, CWE-1230, CWE-1236, CWE-1241, CWE-1284, CWE-1321, CWE-1333, CWE-1336, NVD-CWE-noinfo, NVD-CWE-Other |
| PrimeVul | CWE-17, CWE-20, CWE-22, CWE-59, CWE-79, CWE-94, CWE-119, CWE-120, CWE-122, CWE-125, CWE-134, CWE-189, CWE-190, CWE-191, CWE-193, CWE-200, CWE-212, CWE-252, CWE-264, CWE-269, CWE-275, CWE-276, CWE-284, CWE-287, CWE-288, CWE-295, CWE-310, CWE-327, CWE-345, CWE-354, CWE-362, CWE-369, CWE-399, CWE-400, CWE-401, CWE-415, CWE-416, CWE-434, CWE-444, CWE-476, CWE-522, CWE-552, CWE-617, CWE-665, CWE-668, CWE-672, CWE-703, CWE-704, CWE-732, CWE-754, CWE-770, CWE-772, CWE-787, CWE-824, CWE-834, CWE-835, CWE-843, CWE-862, CWE-863, CWE-908, CWE-909, CWE-924 |
| ReposVul | CWE-17, CWE-20, CWE-22, CWE-59, CWE-74, CWE-78, CWE-79, CWE-89, CWE-94, CWE-119, CWE-120, CWE-121, CWE-122, CWE-125, CWE-134, CWE-189, CWE-190, CWE-191, CWE-193, CWE-200, CWE-203, CWE-212, CWE-252, CWE-264, CWE-269, CWE-275, CWE-276, CWE-284, CWE-287, CWE-288, CWE-295, CWE-310, CWE-327, CWE-345, CWE-354, CWE-362, CWE-369, CWE-399, CWE-400, CWE-401, CWE-415, CWE-416, CWE-434, CWE-444, CWE-476, CWE-522, CWE-552, CWE-617, CWE-665, CWE-668, CWE-672, CWE-703, CWE-704, CWE-732, CWE-754, CWE-763, CWE-770, CWE-772, CWE-787, CWE-823, CWE-824, CWE-834, CWE-835, CWE-843, CWE-862, CWE-863, CWE-908, CWE-909, CWE-924, NVD-CWE-noinfo, NVD-CWE-Other |

### B.9. Case Study

To illustrate the challenges of long-context code security analysis and the importance of cross-function reasoning, we present a case study drawn from the PreciseBugs benchmark. We examine a pair of publicly available implementations—a vulnerable version (vul) and its corresponding patched version (sec)—which exemplify a CWE-20 (Improper Input Validation) vulnerability. Our proposed model correctly identifies the vulnerability, whereas the baseline methods fail to detect it.

**Buggy Version**

In the vulnerable version, early validation failures jump to `chpwfail`, which sends error responses:

```
// Packet length validation
if (req->length < 4) {
    ret = KRB5KRB_AP_ERR_MODIFIED;
    numresult = KRB5_KPASSWD_MALFORMED;
    strlcpy(strresult, "Request was truncated", sizeof(strresult));
    goto chpwfail;  // BUG: Sends error response to unvalidated source
}
```

*Listing 1.* Vulnerable Code - Packet Length Validation

```
// Protocol version check
if (vno != 1 && vno != RFC3244_VERSION) {
    ret = KRB5KDC_ERR_BAD_PVNO;
    numresult = KRB5_KPASSWD_BAD_VERSION;
    snprintf(strresult, sizeof(strresult),
            "Request contained unknown protocol version number %d", vno);
    goto chpwfail;  // BUG: Sends error response to unvalidated source
}
```

*Listing 2.* Vulnerable Code - Protocol Version Check

```
// AP-REQ length validation
if (ptr + ap_req.length > req->data + req->length) {
    ret = KRB5KRB_AP_ERR_MODIFIED;
    numresult = KRB5_KPASSWD_MALFORMED;
    strlcpy(strresult, "Request was truncated in AP-REQ",
            sizeof(strresult));
    goto chpwfail;  // BUG: Sends error response to unvalidated source
}
```

*Listing 3.* Vulnerable Code - AP-REQ Length Validation

The `chpwfail` label constructs and sends an error response:

```
chpwfail:
    ret = alloc_data(&clear, 2 + strlen(strresult));
    if (ret)
        goto bailout;
    ptr = clear.data;
    *ptr++ = (numresult>>8) & 0xff;
    *ptr++ = numresult & 0xff;
    memcpy(ptr, strresult, strlen(strresult));
    // ... constructs and sends error response packet
```

*Listing 4.* The chpwfail Error Handler

**Fixed Version**

The fix changes four instances of `goto chpwfail` to `goto bailout`:

```
// Packet length validation
if (req->length < 4) {
    ret = KRB5KRB_AP_ERR_MODIFIED;
```

```
4    numresult = KRB5_KPASSWD_MALFORMED;
5    strlcpy(strresult, "Request was truncated", sizeof(strresult));
6    goto bailout;  // FIX: Silently drops packet, no response
7 }
```

*Listing 5.* Fixed Code - Silent Packet Drop

The `bailout` label performs cleanup without sending any response:

```
1 bailout:
2     krb5_free_principal(context, changepw);
3     krb5_free_ticket(context, ticket);
4     krb5_auth_con_free(context, auth_context);
5     // ... cleanup only, no packet sent
```

*Listing 6.* The bailout Cleanup Handler

**Root Cause Analysis.** The vulnerability stems from the service's response behavior during early packet validation. When receiving malformed packets, the code would jump to an error-handling routine (`chpwfail`) that constructs and sends an error response. This is problematic because:

1. **No source validation**: The service responds to packets without verifying the legitimacy of the source address

2. **Error responses to invalid input**: Malformed packets receive error responses, which can themselves be interpreted as requests by other services

3. **UDP spoofing**: UDP's connectionless nature allows attackers to forge source addresses

**Why Static Analysis Tools Struggle.** CodeQL and similar SAST tools face difficulties because:

1. **The validation logic is correct** — The code properly checks packet lengths, version numbers, and structure. The bug isn't in *what* is validated but in *what happens after* validation fails.

2. **No obvious dangerous function calls** — Unlike buffer overflows or SQL injection, there's no single "dangerous" function to flag.

3. **Context-dependent semantics** — The difference between `chpwfail` and `bailout` is only meaningful in the context of UDP services and amplification attacks.

4. **Cross-function analysis required** — Understanding requires tracing what `chpwfail` does (sends a packet) versus `bailout` (cleanup only).

