# OpenReview forum: "SecCodePRM: A Process Reward Model for Code Security"
_ICML.cc/2026/Conference — ICML 2026 regular_

### Official Review · Reviewer_JqZp · 2026-02-24

**Soundness:** 2
**Presentation:** 3
**Significance:** 2
**Originality:** 3
**Overall Recommendation:** 4
**Confidence:** 4

**Summary:**

This paper introduces SecCodePRM, a process reward model that assigns step-level security scores during code generation. Trained with labels from static analyzers and expert annotations, it detects fine-grained vulnerabilities in both full and partial code. The model supports three tasks: full-code VD, partial-code VD, and secure code generation. Experiments show consistent gains over prior methods.

**Compliance With Llm Reviewing Policy:**

Affirmed.

**Key Questions For Authors:**

1.  The paper criticizes static analyzers for their inability to handle partial code, yet it relies on tools such as CodeQL to generate step-level reward signals for training. This raises a concern: if static analyzers require complete program context, how can they produce reliable and correct supervision for partial code prefixes? The authors need to clarify (i) which specific CodeQL packages or queries were used, (ii) the accuracy of the resulting labels (e.g., precision/recall against a ground-truth benchmark), and (iii) how human experts were involved to validate or correct the automatically generated labels.
2.  Although the empirical results are promising, the evaluation would be strengthened by comparing against recent agent-based systems (e.g., Claude Code, Copilot) that are increasingly used in interactive coding scenarios. Moreover, because CodeQL itself is used to generate training labels, there is a risk of data leakage. The authors should consider designing an additional experiment with a strict temporal cutoff (e.g., collecting test data from after a certain date) to demonstrate that the model generalizes to unseen vulnerabilities and does not rely on leakage.
3.  Section 5.2 lacks a clear description of how the partial-code samples were constructed. Furthermore, the observed improvement of SecCodePRM on partial-code vulnerability detection is noticeably smaller than its gain on full-code samples. The authors should provide an explanation for this discrepancy. In addition, the baselines chosen for partial-code VD do not include state-of-the-art models or recent agent-based approaches, which may underrepresent the true difficulty of the task.
4.  The claim in the Introduction that “LLM/GNN-based detectors are trained with coarse, program-level binary supervision over full code, which provides limited pressure to attend to fine-grained vulnerable cross-functional/cross-file regions” is somewhat imprecise. Several existing LLM-based approaches can already perform fine-grained vulnerability detection and try to address cross-functional/cross-file contexts. The authors should refine this statement to more accurately reflect the limitations of prior work (e.g., limited detection precision or recall) and support it with appropriate citations.

**Limitations:**

Yes

**Strengths And Weaknesses:**

Strengths:
1. The focus on partial-code vulnerability detection addresses a real need for real-time security in interactive coding, where full context is often unavailable.
2. Adapting process reward models to code security is a novel and well-executed idea.

Weaknesses:
1.  The paper criticizes static analyzers for their inability to handle partial code, yet it relies on tools such as CodeQL to generate step-level reward signals for training. This raises a concern: if static analyzers require complete program context, how can they produce reliable and correct supervision for partial code prefixes? The authors need to clarify (i) which specific CodeQL packages or queries were used, (ii) the accuracy of the resulting labels (e.g., precision/recall against a ground-truth benchmark), and (iii) how human experts were involved to validate or correct the automatically generated labels.
2.  Although the empirical results are promising, the evaluation would be strengthened by comparing against recent agent-based systems (e.g., Claude Code, Copilot) that are increasingly used in interactive coding scenarios. Moreover, because CodeQL itself is used to generate training labels, there is a risk of data leakage. The authors should consider designing an additional experiment with a strict temporal cutoff (e.g., collecting test data from after a certain date) to demonstrate that the model generalizes to unseen vulnerabilities and does not rely on leakage.
3.  Section 5.2 lacks a clear description of how the partial-code samples were constructed. Furthermore, the observed improvement of SecCodePRM on partial-code vulnerability detection is noticeably smaller than its gain on full-code samples. The authors should provide an explanation for this discrepancy. In addition, the baselines chosen for partial-code VD do not include state-of-the-art models or recent agent-based approaches, which may underrepresent the true difficulty of the task.
4.  The claim in the Introduction that “LLM/GNN-based detectors are trained with coarse, program-level binary supervision over full code, which provides limited pressure to attend to fine-grained vulnerable cross-functional/cross-file regions” is somewhat imprecise. Several existing LLM-based approaches can already perform fine-grained vulnerability detection and try to address cross-functional/cross-file contexts. The authors should refine this statement to more accurately reflect the limitations of prior work (e.g., limited detection precision or recall) and support it with appropriate citations.

---

> ### Author Rebuttal · Authors · 2026-03-31
>
> We sincerely thank the reviewer for the valuable feedback.
>
> ### Q1 & W1 CodeQL
>
> We would like to clarify that CodeQL is **not** used to generate step-level supervision in our method. The step-level labels are derived from the line-level ground-truth annotations in the dataset, as in fig.4, rather than from a static analyzer applied to partial prefixes.
>
> In this work, CodeQL is used only for evaluation on the SVEN secure code generation benchmark. For that benchmark, the SVEN authors manually curated the test cases and selected examples that are comparatively robust to CodeQL-based evaluation, which helps reduce the risk that the reported security results are driven by analyzer artifacts rather than genuine vulnerability patterns.
>
> ### Q2 & W2: Claude Code interactive coding agent
>
> Currently, there lacks opensourced agent systems for vulnerability detection. LLMxCPG may be considered as an agent system since it uses two 32B LLMs and tool calling. However, due to the fact that the tool requires full code context, it again can’t address partial code VD.
>
> For closed-source agent systems, we add experiments using Claude Agent SDK, and here’s the results of partial code VD on PrimeVul.
>
> | Model / Method | Acc | F1 |
> |---|---|---|
> | Claude Agent with claude-3-5-sonnet| 53.79 | 63.45|
>
> ### Q3 & W3: How were partial-code samples constructed, and why is the gain smaller than for full-code VD?
>
> We include details on partial-code samples construction in Section.4 and Fig.3. The results on partial code VD are in tab.9. The gain is actually much larger than full-code VD.
>
>
> ### Q4 & W4. The introduction overstates the limitations of prior fine-grained / cross-functional methods.
>
> We would refine this statement to more accurately reflect the limitations of prior work (e.g., limited detection precision or recall) and support it with appropriate citations.

---

> > ### Author Rebuttal · Reviewer_JqZp · 2026-04-01
> >
> > Thanks for your authors' effort and response.

---

### Official Review · Reviewer_uekt · 2026-02-25

**Soundness:** 4
**Presentation:** 3
**Significance:** 3
**Originality:** 2
**Overall Recommendation:** 4
**Confidence:** 2

**Summary:**

This paper proposes a Process Reward Model that can give fine-grained feedback on the safety of (partial) code fragments. Its ability to give immediate feedback is useful for safety constrained generation. To create labeled data, the authors design a pipeline that decomposes coarse-grained vulnerability annotations by comparing vulnerabilities with their fixes. The paper reports improvements over the SOTA on vulnerability detection and safety-constrained generation.

**Compliance With Llm Reviewing Policy:**

Affirmed.

**Final Justification:**

The authors adequately clarified the aspects that were unclear to me and/or missing in the paper. Because the authors committed to update their paper accordingly, I have raised my "Presentation" score to 3 and Overall score to "4 - Weak accept".

**Strengths**:
- The authors show the benefit of PRM models across different benchmarks and include relevant additional analysis (e.g., analyzing effect of code length on performance, analyzing potential side effects of safety-guided code generation).
- Most parts of the paper are clearly written and are supported by useful illustrations. The parts that were unclear or missing have been clarified in the rebuttal and the authors promised to update the paper accordingly.
- Security of code generation models is a very relevant problem. The authors show that process reward models are useful in this context.

**Weaknesses**:
- The paper studies a setting where code is generated in sequential order - code after the cursor position is not included in the context. This is not a representative setup for interactive coding. I do appreciate that the authors will mention this limitation in the paper.
- That fine-grained feedback provided by a PRM is also beneficial in the context of code safety is not that surprising. While the data construction pipeline is very sensible, it didn't involve steps that were particularly innovative.

**Key Questions For Authors:**

1. Could you clarify the inputs of the PRM model?
    - What does the formatted prompt P look like?
    - How do you represent the existing code context?
    - Do you assume there is no code context after the "cursor/generation" position? If yes, I think it would be useful to have some discussion on this (e.g., Can the approach be extended for PRMs for code editing?)

2. Can you include a reference to the "QwenPRM" model? Does this refer to Qwen2.5-Math-PRM-7B? If yes, why do you present this as a general-purpose PRM ?

3. On line 269 it is explained that you introduce errors in the transitive closure of the root cause. Can't this result in ill-defined training examples when the root cause is not part of the context (i.e., when the function with the root cause is occurs after a function in the transitive closure)?

4. Do you have any insights on whether a PRM model would also be useful for RL safety training?

**Limitations:**

yes

**Strengths And Weaknesses:**

**Strengths**:
- [Soundness] All claims are supported with references or experiments. The authors show the benefit of PRM models across different benchmarks and include relevant additional analysis (e.g., analyzing effect of code length on performance, analyzing potential side effects of safety-guided code generation).
- [Presentation] Most parts of the paper are clearly written and are supported by useful illustrations.
- [Significance] Security of code generation models is a very relevant problem. The authors show that process reward models are useful in this context.

**Weaknesses**:
- [Presentation] Some key aspects are unclear:
  -The paper mentions "full code" (e.g., l128 left column) and "complete code sequences" (e.g., l141, right column) but I didn't find precise definition for these terms. Do they refer to a complete function body, a complete module, an entire repository? Such terms are also used when discussing the granularity of SotA methods, which is therefore also unclear.
  -The inputs of the PRM model are not clearly described:
	- In Section 3.2, it's unclear to me how the existing code context fits in. Based on the current explanation I believe the authors implicitly assume that code is generated in the order of the final source code (line n of a file is always coded before line n+1). As such, the code context for step s_t is defined by the previous trajectory s_0, … s_{t-1}.
	- The Dataset format paragraph in Section 4 (l247-254, right column) seems inconsistent with Section 3.2: it uses different symbols and mentions "a formatted prompt P in natural language". It's also unclear what this formatted prompt P exactly entails (Does this encode a user intent? Is it a system prompt?).
	- Appendix B.6 includes a prompt template for the vulnerability detection baselines, but I didn't find a prompt template for the PRM Model.
- [Presentation/Soundness] In Section 5.3 on secure code generation I'm not sure which model "QwenPRM" refers to. The authors claim this is a general-purpose PRM model, but the only Qwen PRMs I can find online are Qwen2.5-Math-PRM-7B and Qwen2.5-Math-PRM-72B (which specialize in mathematics).
- [Significance] The paper seems to study a setting where code is generated in sequential order (see earlier remarks). This is not a representative setup for interactive coding.
- [Originality] That fine-grained feedback provided by a PRM is also beneficial in the context of code safety is not that surprising. While the data construction pipeline is very sensible, it didn't require steps that were particularly innovative.

Minor comments:
- Include citations for the other LLMs as well
- There is often a space missing between SecCodePRM and the next word.

Overall, I think this paper has promise, but its presentation should be improved.

---

> ### Author Rebuttal · Authors · 2026-03-31
>
> We sincerely thank the reviewer for the valuable feedback.
>
> ### Q1 &W1. Clarification on the inputs to the PRM model
>
> We thank the reviewer for pointing out that the PRM input format was not described clearly enough.
>
> 1. **Prompt format.**
>    The PRM input uses a natural-language instruction of the form:
>    *“Given the previous code ..., determine whether the current code ... is vulnerable or not. Reason step by step, and answer with Yes or No. Put your answer in {}.”*
>
> 2. **How existing code context is incorporated.**
>    The existing code context is included directly in the prompt as the **previous code**. That is, the model conditions on the available prefix/context exactly as written, without requiring any special transformation beyond insertion into the prompt template.
>
> 3. **Assumption about future code context.**
>    Yes, we assume there is **no code context after the current cursor/generation position**. SecCodePRM is designed for **prefix-based vulnerability detection during generation**, rather than code editing with both left and right context. Within this setting, the model can still assign vulnerability labels to individual steps in the observed code context.
>
> ### Q2 & W2. Clarification on the reference to “QwenPRM”
>
> We thank the reviewer for pointing this out. “QwenPRM” refers specifically to **Qwen/Qwen2.5-Math-PRM-7B**, and we will revise the paper to state this explicitly.
>
> We use this model as a representative example of a **general-purpose process reward model** and we aim to emphasize it's not a specially trained security-purpose PRM. Prior work has shown that such PRMs, although not trained specifically for code security, can still improve coding performance or test-time scaling in code-related settings [1,2]. We will clarify this point and provide the full model name in the revision.
>
> [1] *From Mathematical Reasoning to Code: Generalization of Process Reward Models in Test-Time Scaling.*
> [2] *Process Reward Models That Think (ThinkPRM).*
>
> ### Q3. Clarification on transitive closure
>
> Please see our response to **Reviewer NzdJ, Weakness 3**, where we provide results for the setting in which SecCodePRM is **trained with transitive-closure labels but evaluated without transitive closure**. This directly addresses the concern that the model’s performance may depend on the propagation heuristic.
>
> ### Q4. Whether a PRM would also be useful for RL-based safety training
>
> Yes. A PRM can provide **dense intermediate rewards**, rather than relying only on sparse end-of-trajectory rewards. This is especially valuable for code generation, where trajectories are long and a final security signal may arrive too late to guide learning effectively. In such settings, a PRM can assign training signals to **partial rollouts**, enabling more informative and potentially more efficient RL optimization.
>
> ### W1. Clarification of “full code,” “complete code sequences,” and how existing context is used
>
> We thank the reviewer for pointing out that these terms were not defined clearly enough.
>
> 1. **“Full code” / “complete code sequences.”**
>    By *full code*, we mean code provided with the **full available context**, such as necessary imports, surrounding definitions, and the complete realization of the target function or program unit being analyzed. Similarly, *complete code sequences* refers to code that is fully written for the scope of the task, rather than an intermediate prefix or partial trajectory.
>
> 2. **How the existing code context fits in.**
>    For generation tasks, the existing code context is included directly in the model input as the observed prefix. SecCodePRM then assigns **per-step security scores** to candidate continuations and uses these scores to rank and select among candidates during inference.

---

> > ### Author Rebuttal · Reviewer_uekt · 2026-04-01
> >
> > I have read the rebuttal. The authors adequately clarified the aspects that were unclear to me and/or missing in the paper.  Because the authors committed to update their paper accordingly, I will raise my "Presentation" score to 3 and Overall score to "4 - Weak accept".
> >
> > I am not inclined to go higher because:
> > * The results are not particularly surprising in my opinion;
> > * The method does not take into account code after the cursor position, which seems suboptimal for the generation use cases.
> >
> > Minor remark: I think it is imprecise to refer to **Qwen/Qwen2.5-Math-PRM-7B** as a general purpose PRM, even when considering it has been proven useful in coding contexts.

---

> > > ### Author Response · Authors · 2026-04-02
> > >
> > > We sincerely thank the reviewer for the thoughtful feedback. In the final version, we will revise the phrase “general-purpose PRM”.
> > >
> > > Regarding the comment that the results are not particularly surprising:
> > >
> > > We respectfully believe the contribution of this work extends beyond the absolute performance gains on full-code vulnerability detection, partial-code vulnerability detection, and secure code generation. Specifically, to the best of our knowledge, prior work has not introduced a model explicitly trained to handle the partial-code vulnerability detection setting. Our work therefore addresses an important gap in the literature. In addition, existing approaches typically require separate models tailored to individual security tasks, whereas our framework provides a unified model that can be applied across all three settings. We believe this task unification is practically important and constitutes a substantive contribution beyond the empirical improvements alone.
> > >
> > > Regarding the concern that the method does not use code after the cursor position:
> > >
> > > We agree this is an important direction. At present, however, there is a lack of security-focused benchmarks and training data for this setting, which makes rigorous study difficult. We will clarify this limitation in the paper. We also view this as a promising avenue for future work, including the construction of benchmarks and corresponding training datasets that incorporate post-cursor context for security-aware code modeling.

---

### Official Review · Reviewer_NzdJ · 2026-03-11

**Soundness:** 3
**Presentation:** 2
**Significance:** 3
**Originality:** 3
**Overall Recommendation:** 3
**Confidence:** 4

**Summary:**

The paper proposes SecCodePRM, a step-wise reward modeling framework trained using supervision derived from static analyzers and expert annotations. The method is evaluated on multiple vulnerability detection benchmarks (e.g., SVEN, PrimeVul, PreciseBugs) and is also used to guide secure code generation via inference-time scaling. Empirical results show strong improvements over prior baselines across several tasks.

The idea of shifting from global, full-completion evaluation to partial-code, process-level security modeling is interesting and potentially impactful. However, several technical and methodological issues limit the strength of the current submission.

**Compliance With Llm Reviewing Policy:**

Affirmed.

**Key Questions For Authors:**

See my concerns

**Limitations:**

Yes

**Strengths And Weaknesses:**

1. The paper identifies an important gap between full-program supervision and real-time or partial-code vulnerability detection.

2. The same PRM is used for full-code detection, partial-code detection, and secure code generation, which is conceptually appealing.

3. The reported results show substantial improvements over baselines across multiple datasets, particularly in pairwise PrimeVul metrics and partial-code settings.

## Major Concerns

1. In the introduction, the authors state that prior methods “provide limited pressure to attend to fine-grained vulnerable cross-functional/cross-file regions.” However:
The training setup does not explicitly incorporate a mechanism specifically designed for cross-file reasoning.
The experimental section does not provide a clearly verifiable experimental design targeting cross-file vulnerability detection.

2. The introduction argues that the method enables “more efficient optimization in settings such as reinforcement learning, where full-rollout reward computation becomes increasingly expensive.” However:
No reinforcement learning experiment is presented.
There is no quantitative comparison of rollout cost, reward sparsity, or optimization efficiency.
Thus, the RL efficiency argument remains conceptual and is not experimentally validated in the paper.

3. The label propagation strategy assigns negative labels to caller functions via AST-based transitive closure, assuming that a vulnerability at a callee function corrupts all callers.
However, this may introduce noise:
A vulnerability may only occur along specific execution paths.
A caller may not actually execute the vulnerable path.
Treating all caller steps as ground-truth negative may mislabel safe contexts.
Since this propagation strategy appears to be used both during training and evaluation, it raises concerns about evaluation correctness.
If the same heuristic shapes both training and test labels, the reported performance may partially reflect consistency with the heuristic rather than true vulnerability understanding.

A useful additional experiment would be:
Train with propagation,
Evaluate without propagation (i.e., on strictly localized ground-truth labels),
to demonstrate robustness beyond heuristic alignment.

4. While the paper demonstrates strong performance of SecCodePRM, it does not fully eliminate the concern that improvements may stem from task-specific exposure during training rather than from the intrinsic advantage of process-level supervision.

In particular, the paper does not clearly compare:
A Process Reward Model (PRM), and
A Final Reward Model (FRM) trained with only full-trajectory supervision,
under otherwise controlled conditions.
Without such an ablation, it remains unclear whether the gains arise specifically from step-level supervision, or simply from additional training on vulnerability-labeled data.

I strongly recommend adding an ablation study comparing PRM and FRM to more rigorously demonstrate the unique contribution of process-level reward modeling.

5. I did not fully understand the split of the training/testing set. It seems that the authors use both SVEN/PrimeVul as training and testing dataset. Do the authors follow the default split of those dataset?

6. That is weired that the authors compare with different tools on different dataset. It would be better if the authors can present comparision results with the same training/testing set among multiple tools.

7. This paper is not easy to follow. There are a lot of grammer issues. And, there are multiple formatting and naming inconsistencies that should be corrected:
- Line 52: “SecCodePRMcan tackle both complete and partial code vulnerability detection” — missing space between “SecCodePRM” and “can”.
- Line 107: “The proposed SecCodePRMconsistently improves performance…” — missing space.
- Line 140: “The SecCodePRMoperates at the granularity…” — missing space.
- Line 210: “The SecCodePRMarchitecture offers several key advantages” — missing space.
- Line 216: “SecCodePRMis flexible enough…” — missing space.
In the introduction, the method name is written as “SecCoderPRM” instead of “SecCodePRM”.

---

> ### Author Rebuttal · Authors · 2026-03-31
>
> We sincerely thank the reviewer for the valuable feedback.
>
> ### W1 and W2: explicitly incorporate a mechanism specifically designed for cross-file reasoning. RL efficiency claim.
>
> We will narrow the wording in the introduction.
>
> ### W3: Train with propagation, Evaluate without propagation.
>
> This does not affect the full VD, so we add experiments for partial VD here, the performance remains highly consistent under both evaluation protocols, with only negligible differences overall, and is even slightly higher on some datasets. In particular, evaluation on ground-truth labels yields comparable results to evaluation with propagation on **SVEN** and **PrimeVul**, while achieving slightly better performance on **ReposVul** and **PreciseBugs**. This suggests that SecCodePRM’s gains do not depend on the propagation heuristic and remain robust under stricter ground-truth evaluation. We will add in Tab.9.
>
> | Model | SVEN F1 | SVEN Acc | PrimeVul F1 | PrimeVul Acc | ReposVul F1 | ReposVul Acc | PreciseBugs F1 | PreciseBugs Acc |
> |---|---|---|---|---|---|---|---|---|
> | Eval on propagation | 50.17 | 93.45 | 55.71 | 92.57 | 23.79 | 99.33 | 90.00 | 93.94 |
> | Eval on ground truth | 49.52 | 92.85 | 55.33 | 92.52 | 24.36 | 99.35 | 92.86 | 94.41 |
>
> ### W4. PRM vs FRM ablation.
>
> We agree that this is an important ablation for full VD. SecCodePRM substantially outperforms the FRM baseline by a large margin on both metrics. Note that SecCodePRM could perform partial VD while FRM can't. We will add this in Tab.3.
> | Model / Method | Acc | F1↑|
> |---|---|---|
> | FRM |72.28 |   61.65 |
> | *SecCodePRM* | **96.83** | **96.73** |
>
> ### W5. Do you follow the default train/test splits of SVEN and PrimeVul?
>
> Yes. We follow the default train/test splits provided by those datasets, and the tab. 11 shows the statistics of train and test split.
>
> ### W6. Comparisons are shown with different tools on different datasets.
>
> We will clarify this evaluation setup explicitly in the appendix. The choice of evaluation protocol depends on the benchmark rather than on our method:
>
> - **Full-code vulnerability detection:** no external tools are used; evaluation is based on ground-truth labels.
> - **Partial-code vulnerability detection:** no external tools are used; evaluation is based on ground-truth labels.
> - **Secure code generation:** on **SVEN**, security is evaluated with **CodeQL**; on **CWEval**, security is evaluated with **unit tests** following the benchmark setup.
> - **General code generation:** on **LiveCodeBench**, evaluation is based on **unit-test passing**.
>
> ### W7. Grammar / formatting / naming inconsistencies
>
> Thank you. We have identified and will fix the formatting issues, typos, and naming inconsistencies (including spacing and model naming).

---

> > ### Author Rebuttal · Reviewer_NzdJ · 2026-04-02
> >
> > Thank the authors for their efforts.

---

> > > ### Author Response · Authors · 2026-04-02
> > >
> > > We sincerely thank the reviewer for the valuable feedback. We do think the FRM vs PRM, and the evaluation on ground truth ablation are very insightful. We promise to include all the responses here in the final version. Since all the questions are fully resolved, could the reviewer consider adjusting the overall score?

---

### Official Review · Reviewer_s3QC · 2026-03-19

**Soundness:** 2
**Presentation:** 3
**Significance:** 4
**Originality:** 4
**Overall Recommendation:** 2
**Confidence:** 4

**Summary:**

The paper introduces SecCodePRM, a Process Reward Model for detecting security vulnerabilities in both full and partial source code. The source code is divided into "steps" through delimiters like \n\n (and some additional heuristic merging logic). The model, initialized with a decoder-only pre-trained LM, takes source code as input and learns to predict a vulnerability score for each step. The step-level vulnerability scores are aggregated using a risk-sensitive weighting to yield an overall score for the given source code. The paper describes methods for extracting granular supervision signals to learn these step-classifiers, either from vulnerable-secured pairs using sequence alignment or from unpaired data. In experiments, the authors claim significant gains over off-the-shelf LLMs and prior LLM+Graph-NN based approaches. Using these step-level classifiers, the authors show that they can also steer the decoding process via inference-time scaling to generate more secure code without sacrificing functional correctness. Overall, I enjoyed reading this paper but I think the current version requires significant improvements.

**Compliance With Llm Reviewing Policy:**

Affirmed.

**Key Questions For Authors:**

* How do standard security-focused reward models fail when applied to partial code, and how does your method help over these existing models? A comparison would be helpful.

* How does SecCodePRM compares with CodeQL for fully-written code?

* It is unclear how advantages defined in Line 190 are finally used during decoding.

* Birdirectional implementation: Is it possible to construct SecCodePRM with bidirectional attention?  Given that vulnerability confirmation often requires checking cross-file or cross-function interactions, could a bidirectional attention mechanism perform better than a strictly causal one? How do you reconcile the use of full-code ASTs for label propagation with the strictly autoregressive, prefix-only nature of the PRM during training and inference?

* Figure 5 effectively shows SecCodePRM's accuracy across token lengths, but plotting a baseline method's degradation on the same chart would significantly strengthen your claim regarding length resilience.

*

**Limitations:**

yes

**Strengths And Weaknesses:**

## Strengths
* Using step-level scores to predict vulnerability at the prefix level and guiding the decoding process to generate secure code is a smart idea. Partial Code Evaluation is a problem of practical significance.
* The empirical demonstration that SecCodePRM maintains accuracy across significantly long context windows is a strong structural advantage.
* There are plenty of experiments in the paper (however, I have some concerns elaborated in the weaknesses described below). It is nice to see authors evaluate both functional correctness and security (e.g., using CWEval) demonstrating that the proposed approach does not degrade general coding capabilities.

## Weaknesses
* **Weak Baselines / Evals**: Table 2 compares SecCodePRM against relatively old models (CodeT5, CodeBERT, CodeGen2.5). Furthermore, the methodology for constructing the StarCoder2 baseline is not clear -- unclear if SecCodePRM was applied on top of it or how the evaluation was structured. How would SecCodePRM perform on more recent benchmarks like Bounty-Bench: https://arxiv.org/abs/2505.15216. Also, it would be nice to have numbers for more recent models like Claude, Qwen3Coder, Minimax, GPT-5 (not all of these, but some of these).

* Incorrect Supervision / Train-test mismatch in how supervision is derived: The data construction pipeline relies on full-code context and Abstract Syntax Trees (ASTs) to retroactively propagate vulnerability labels to caller functions. However, the model evaluates code prefixes autoregressively. For example, if a sequence unfolds as step-1, step-2, caller-1, ... [future steps] ... first failure, the AST will label caller-1 as vulnerable. Yet, an autoregressive model lacks the future context to know caller-1 will invoke a vulnerable sink until that "first failure" is actually generated. This forces a strictly causal architecture to predict bidirectional dependencies, resulting in flawed and noisy supervision.


## Minor:
* Missing reference on Line 157 (For Example, ...)
* Line 165 -- should this be called as probability margin (and not logit margin)
* Experiments: Line 300 -- No context provided for what is Detection Threshold, Scaling Strategy, ...
* The paper frequently mentions "inference-time scaling" and "Advantage-based Selection" but lacks clarity on the exact decoding mechanism. It is unclear if the authors are employing standard Best-of-N sampling, beam search, or a more complex tree-search variant.

---

> ### Author Rebuttal · Authors · 2026-03-31
>
> We sincerely thank the reviewer for the valuable feedback.
>
> ### W1: 1) old models in tab.2; 2) StarCoder2 baseline construction; 3) Bounty-Bench.
>
> 1) We have added a stronger reference point in Table 2 with a more recent model:
> | Model / Method | PC↑ | PV↓ | PB↓ | PR↓ |
> |---|---|---|---|---|
> | GPT5.4| 0.1281 | 0.4975 | 0.3399 | 0.0345 |
>
> 2) SecCodePRM is not built on StarCoder and does not share its backbone. Our model is built on Qwen2.5-Coder-7B. For the StarCoder2 baseline, we follow the same evaluation protocol used in PrimeVul.
>
> 3) At present, Bounty-Bench does not support evaluation with local models and only supports API-based models through providers such as Anthropic and Together AI. We therefore could not evaluate at this time, but we will add those results once local-model evaluation becomes available.
>
> ### W2: AST and autoregressive model.
>
> Our propagation rule does not label earlier caller steps as vulnerable. Concretely, if step \(s_k\) is identified as the root vulnerable step, we only propagate vulnerability labels to caller/function-context steps \(s_j\) with \(j > k\), i.e., steps that appear **after** the root cause has already been introduced in the observed trajectory. Therefore, the model is never asked to predict a vulnerability based on unavailable future evidence. In this sense, the supervision remains consistent with the causal, prefix-based setting. We could clarify this in sec.4.
>
> ### Q1. How do existing security-focused reward models perform on partial code, and how does our method improve upon them?
>
> Current security-focused reward models are
>
> 1. **Full-code detectors and static-analysis-based pipelines require near-complete context.**
>    State-of-the-art full-code detectors such as **LLMxCPG** rely on static analyzers in the pipeline, which generally require substantially complete code and therefore do not support partial-code evaluation. Other strong baselines, such as **ReGVD**, depend on graph-based representations derived from full programs; as a result, they are also designed around substantially complete context rather than prefixes.
>
> 2. **General-purpose coding reward models** are not trained for security-sensitive step-level supervision.
>
> SecCodePRM is designed to fill this gap. It is trained with **security-oriented, step-level supervision** and can evaluate **partial code trajectories** directly.
>
> ### Q2. How does SecCodePRM compare with CodeQL on fully written code?
>
> CodeQL shows limited accuracy on repo-level, fully written code. In [1], its accuracy is around **22.5%**. In contrast, SecCodePRM performs strongly on repo-level full-code settings, reaching around **90% accuracy** on the **PreciseBugs**  whose full-code samples have an average length of roughly **19,000 tokens**.
>
> Beyond accuracy, CodeQL typically requires compilation and can be slow and inefficient on long-context codebases [2], while SecCodePRM operates as an end-to-end learned model and can produce predictions within seconds, more suitable for real-time or large-scale settings.
>
> [1] Li, Ziyang, Saikat Dutta, and Mayur Naik. *IRIS: LLM-Assisted Static Analysis for Detecting Security Vulnerabilities.* The Thirteenth ICLR.
> [2] Li, Fengjie, et al. *LLM-based Vulnerability Detection at Project Scale: An Empirical Study.* arXiv preprint arXiv:2601.19239 (2026).
>
> ### Q3. How are the advantages defined in Line 190 used during decoding?
>
> During decoding, the PRM assigns a step reward to each candidate continuation at every step, and the advantage is used as a relative ranking signal among candidates in the current batch. In practice, inference prioritizes continuations with higher **cumulative reweighted reward**.
>
> ### Q4. Could SecCodePRM be implemented with bidirectional attention?
>
> This is a valuable suggestion. In principle, a bidirectional variant could be beneficial for **offline full-code vulnerability analysis**, where the future context can be used directly. However, our backbone is a causal coding model rather than a bidirectional one. Training a strong bidirectional coding LLM from scratch would also be substantially more expensive. We agree, however, that bidirectional SecCodePRM is an interesting direction for future work.
>
> ### Q5. A baseline degradation curve on the same length plot would strengthen the claim.
>
> We agree. In the revision, we will add a directly comparable baseline curve to the length-resilience figure. Since we cannot add a new figure in the rebuttal, we provide representative numbers here: for the baseline method, accuracy drops to **below 60%** for samples longer than **10,000** tokens, and further declines to around **20%** in the **14,000--19,000** token range.
>
> ### M3 Detection Threshold / Scaling Strategy
>
> In the revision, we will explicitly describe the thresholding rule used to convert the aggregated reward into a binary vulnerability decision and clarify that the “scaling strategy” refers to **PRM-guided candidate ranking during inference-time selection**.

---

> > ### Author Rebuttal · Reviewer_s3QC · 2026-04-06
> >
> > I would like to thank the authors for their response. After reviewing the author's response and other reviews, I will maintain my rating, primarily because the baselines and evaluations still need improvement.
> >
> > As a follow-up, in response to reviewer JqZp's concerns, the authors provide the performance of Claude Agent with claude-3-5-sonnet. Could the authors clarify why this baseline is not considered in their response above, which only includes GPT-5.4 numbers for Table 2?

---

### Decision · Program_Chairs · 2026-04-30

**Decision:**

Accept (regular)

**Comment:**

I like the ideas in the paper, especially the emphasis on partial code vulnerability detection -- quite appealing and very practical challenge today. Reviewers agree that this is the main highlight of the paper. Reviewer NzdJ raised quite a few interesting points and concerns, the authors provided a detailed response, which the reviewer acknowledged as "fully resolved" -- but hasn't updated the score. The outstanding issue is the unresolved concerns of the most critical reviewer s3QC, especially the improvements on baselines and evaluations. I'm tending to accept the paper, but it would be good to discuss this.